# EntRAG: Entity-Centric Retrieval-Augmented Generation for Knowledge-based Visual Question Answering

**Yiheng Hu** [1 2]  **Xiaoyang Wang** [1]  **Qing Liu** [1 2]  **Xiwei Xu** [2]  **Qian Fu** [2]  **Wenjie Zhang** [1]

## Abstract

Knowledge-based Visual Question Answering (KB-VQA) remains a challenging task, particularly when queries require precise identification and grounding of fine-grained entities within large-scale knowledge base. Existing methods often treat visual and textual signals in isolation and rely heavily on image-centric retrieval, which makes them sensitive to visual ambiguities. To address these limitations, we propose EntRAG, an entity-centric retrieval-augmented generation framework. Our approach first introduces Ent-Bind to align query representations with multimodal entity embeddings by explicitly binding entity tokens to latent visual features, retrieving a set of relevant candidate entities. A reranking mechanism is applied to these candidate entities to select the most informative context by combining entity-level alignment with overall contextual relevance. The selected evidence is incorporated into context-aware generation module to produce final answer. By explicitly operating at the entity level, EntRAG achieves more consistent and reliable results. Extensive experiments demonstrate that EntRAG consistently outperforms prior methods, achieving scores of 46.1 on E-VQA and 44.5 on InfoSeek.

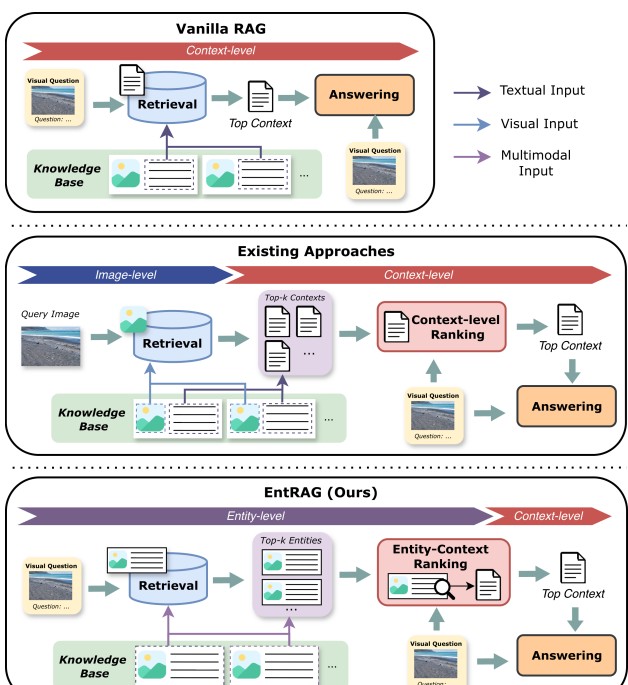

*Figure 1.* Comparison of different frameworks. **Vanilla RAG** operates at the context level, retrieving textual snippets and directly feeding the top-ranked context to the answer generator. **Existing approaches** first perform image-dominant retrieval, then use associated text as top-k contexts, followed by context-level reranking to select evidence. Our approach, **EntRAG**, performs entity-level retrieval with multimodal inputs, explicitly grounding entities before locating and ranking relevant contexts.

## 1. Introduction

Recent advances in multimodal large language models (MLLMs) have substantially improved the performance of Visual Question Answering (VQA), largely due to their strong multimodal encoders and effective alignment between visual representations and large language models

[1]University of New South Wales, Sydney, Australia [2]CSIRO Data61, Australia. Correspondence to: Xiaoyang Wang <xiaoyang.wang1@unsw.edu.au>.

*Proceedings of the 43rd International Conference on Machine Learning*, Seoul, South Korea. PMLR 306, 2026. Copyright 2026 by the author(s).

(LLMs). These models exhibit impressive reasoning capabilities and can often leverage rich parametric knowledge acquired during large-scale pretraining. Within this landscape, knowledge-based visual question answering (KB-VQA) has emerged as an important and challenging subfield of VQA, where question answering requires knowledge beyond visual context. Existing KB-VQA tasks can be broadly categorized into two types. The first type is represented by datasets such as OK-VQA (Marino et al., 2019), which focus on open-domain questions that primarily rely on commonsense or general world knowledge. Recent works (Yang et al., 2025; Cao & Jiang, 2024) demonstrate that MLLMs can achieve strong performance by exploiting the paramet-

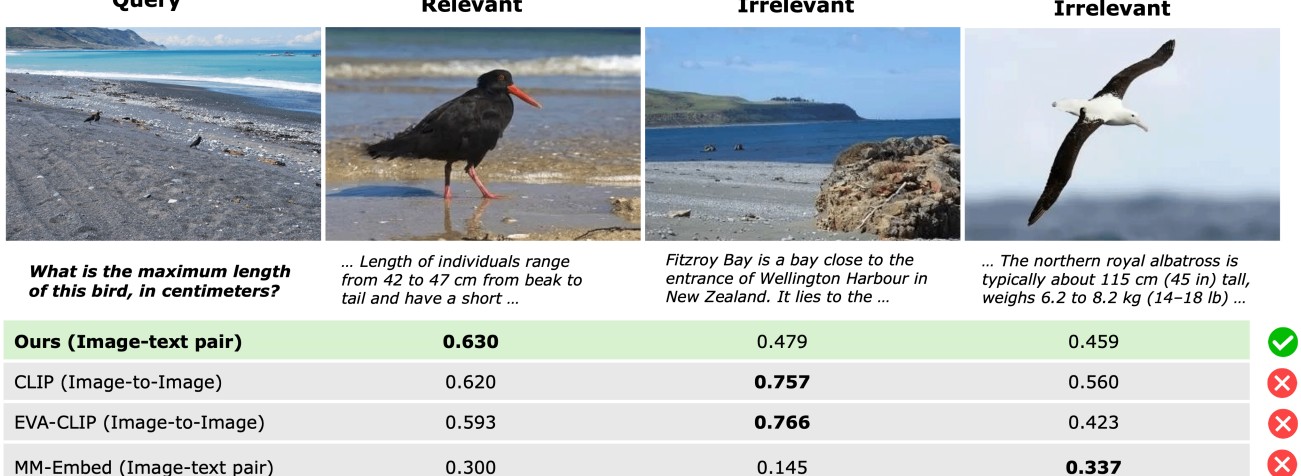

| | Query | Relevant | Irrelevant | Irrelevant | |
|---|---|---|---|---|---|
| | *What is the maximum length of this bird, in centimeters?* | *… Length of individuals range from 42 to 47 cm from beak to tail and have a short …* | *Fitzroy Bay is a bay close to the entrance of Wellington Harbour in New Zealand. It lies to the …* | *… The northern royal albatross is typically about 115 cm (45 in) tall, weighs 6.2 to 8.2 kg (14–18 lb) …* | |
| **Ours (Image-text pair)** | | **0.630** | 0.479 | 0.459 | ✅ |
| CLIP (Image-to-Image) | | 0.620 | **0.757** | 0.560 | ❌ |
| EVA-CLIP (Image-to-Image) | | 0.593 | **0.766** | 0.423 | ❌ |
| MM-Embed (Image-text pair) | | 0.300 | 0.145 | **0.337** | ❌ |

*Figure 2.* Challenges in Multimodal Entity Retrieval. Given a query regarding coastal birds, standard image-to-image retrieval (e.g., CLIP, EVA-CLIP) exhibits a bias toward global scene context (the coastline) over the target entity. While MM-Embed integrates textual cues to focus on the "bird" category, its reliance on coarse linguistic alignment leads to a mismatch with visually similar but incorrect entities. In contrast, our approach effectively integrates fine-grained visual details with textual entity signals to achieve precise grounding.

ric knowledge of large foundation models without explicit access to external knowledge sources. The second type of KB-VQA, exemplified by datasets such as Encyclopedic VQA (E-VQA) and InfoSeek, targets fine-grained entity and often requires specialized or long-tail knowledge. Unlike the first type, these datasets provide a dedicated external knowledge base, requiring models to explicitly retrieve and integrate information from it to produce correct answers.

In this work, we focus on the second category of KB-VQA, which remains challenging as large foundation models often struggle to identify long-tail entities sparsely represented during pretraining (Kandpal et al., 2023). Many prior works (Cocchi et al., 2025; Yan & Xie, 2024) address this via image-based retrieval using CLIP-style encoders, followed by context-level reranking over textual snippets (Figure 1). However, knowledge base entries are inherently multimodal, consisting of paired images and text, and existing pipelines process these modalities independently. This leads to two main challenges: (1) **Modality-Isolated Representation**. By treating images and text separately, retrieval and reranking fail to represent an entity as a unified multimodal concept. Without entity guidance, images lack clear cues for the target, so retrieval relies on global features. Likewise, reranking over textual snippets struggles to distinguish the correct evidence when multiple candidates are equally plausible. (2) **Vulnerability to Visual Ambiguity**. Retrieval can be further impaired in images with cluttered backgrounds or multiple salient objects. For example, in Figure 2, the query targets a bird species, but the bird occupies a small portion of a coastal scene. Visual encoders

(e.g., CLIP, EVA-CLIP) prioritize dominant background features such as the "coastline," overlooking fine-grained cues needed to correctly identify the bird.

To address the aforementioned challenges, we propose EntRAG, a retrieval-augmented generation (RAG) framework for KB-VQA, which progressively grounds entities, locate relevant entity-specific contexts and generates knowledge-conditioned answer. The framework consists of three key components: (i) EntBind for Entity Retrieval. To improve entity identification and tightly integrate visual and textual cues, we introduce a retrieval model that binds entity-level textual and fine-grained visual features into a unified embedding space. Specifically, we introduce an explicit entity token to anchor entity semantics within MLLMs, while visual representations are derived from patch-level features and aggregated via attention pooling conditioned on the entity token. This design enables precise entity-centric matching during the initial retrieval stage. (ii) ECRanker for Entity-Context Reranking. To further refine entity selection and identify relevant contexts, we introduce a re-ranking module that jointly considers entity-level alignment and query–context relevance. These complementary signals allow the ranker to prioritizing knowledge that are both entity-consistent and contextually informative for answering. (iii) CAGen for Context-Aware Answer Generation. Finally, to support accurate reasoning and answer generation, we employ a MLLM that explicitly condition the generation process on retrieved entity-centric knowledge, ensuring faithful utilization of the retrieved evidence.

Our main contributions are summarized as follows:

1. We introduce EntBind that tightly integrate textual and visual signals at the entity level, producing effective multimodal embeddings that better capture fine-grained entity semantics and support accurate entity identification.

2. We introduce EntRAG, a KB-VQA framework that represents entities as unified multimodal concepts instead of separating textual and visual cues. This entity-centric perspective enhances context selection and ensures that retrieved knowledge is more directly relevant to the query.

3. Extensive experiments on KB-VQA benchmarks, E-VQA and Infoseek, demonstrate that our approach achieves state-of-the-art performance.

## 2. Related Works

### 2.1. Multimodal Retrieval

Early KB-VQA approaches primarily retrieved textual passages using multimodal queries (Qu et al., 2021; Lin et al., 2024). While effective in open-domain settings, text-only evidence often lacks the precision needed for fine-grained entity recognition. To leverage visual signals, later methods (Caffagni et al., 2024) employed multimodal encoders like CLIP (Radford et al., 2021), but its coarse visual representations limit accurate entity identification. High-capacity encoders such as EVA-CLIP (Sun et al., 2024) (Cocchi et al., 2025; Yan & Xie, 2024; Gu et al., 2026) preserve fine-grained visual details essential for large-scale entity discrimination. Nevertheless, computing embeddings for entire images can obscure target entities when multiple objects or dominant backgrounds are present (Figure 2). Wiki-PRF (Gu et al., 2026) mitigates this by cropping entity regions before embedding, but relying solely on visual features makes retrieval and reasoning sensitive to cropping or detection errors.

In parallel, MLLMs are increasingly used for multimodal embedding generation (Lin et al., 2025; Jiang et al., 2024; Gu et al., 2025), leveraging large-scale, diverse training (Wei et al., 2024) to perform complex reasoning. However, even when KB-VQA datasets like InfoSeek are included in training, their ability to recognize specific entities remains limited. Figure 2 shows MM-Embed (Lin et al., 2025) correctly identifies a bird query but does not fully exploit its visual appearance to determine the precise species.

### 2.2. Knowledge-based Visual Question Answering

In this section, we focus on the second type of knowledge-based visual question answering introduced in Section 1, which often involves fine-grained entities. To address this task, retrieval-augmented generation (RAG) has emerged as a promising paradigm for KB-VQA. The typical pipeline proceeds in stages: the query image is first used to identify relevant Wikipedia entities, after which candidate textual articles are reranked to select the most informative evidence. Finally, LLMs or MLLMs generate answers grounded in the retrieved knowledge. This retrieve-then-generate paradigm has also been widely adopted in broader LLM reasoning tasks (Tan et al., 2026a;b). Since the initial retrieval stage has been discussed in Section 2.1, here we focus primarily on the subsequent ranking, and answer generation stages.

Wiki-LLaVA (Caffagni et al., 2024) employs a Contriever-based architecture to rerank textual chunks. Similarly, EchoSight (Yan & Xie, 2024) encodes image–text queries with a Q-Former to rank textual candidate evidence. Both methods rely on embedding-based reranking to identify relevant textual knowledge. In contrast, ReflectiVA (Cocchi et al., 2025) introduces reflective tokens to fine-tune models for both snippet reranking and answer generation using an autoregressive objective. Wiki-PRF (Gu et al., 2026) filters reranked knowledge via reinforcement learning to improve answer accuracy. More recently, Wiki-R1 (?) applies curriculum reinforcement learning to improve reasoning robustness over noisy retrieved context, gradually increasing training difficulty by varying the number of retrieved candidates. Despite these advances, all of these approaches focus predominantly on contextual-level information in later stage.

## 3. Problem Setup

### 3.1. Task Definition

Given the question $\mathcal{Q}$ along with image $\mathcal{I}_Q$, and a multimodal knowledge base $\mathcal{K}$. The knowledge base consists of a set of entities $\mathcal{K} = \{e_1, e_2, \cdots, e_N\}$. Each entity $e$ is annotated with multimodal knowledge $e = (\mathcal{I}_e, \mathcal{T}_e)$, where $\mathcal{I}_e = \{I_e^1, \cdots, I_e^{M_e}\}$ is a set of images depicting entity $e$, and $\mathcal{T}_e = \{T_e^1, \cdots, T_e^{L_e}\}$ is a set of textual snippets about entity $e$. The goal of KB-VQA is to generate an answer $\mathcal{A}$ as following:

$$\hat{\mathcal{A}} = \arg\max_{\mathcal{A}} P(\mathcal{A}|\mathcal{I}_Q, \mathcal{Q}, \mathcal{K}). \tag{1}$$

### 3.2. Motivation

We consider the task of KB-VQA, in which a model must identify and utilize evidence from a multimodal knowledge base $\mathcal{K}$ to generate an answer $\mathcal{A}$. Our approach is grounded in the following two assumptions:

**Assumption 3.1** (Information Bottleneck)**.** We posit that the generative process for an answer $\mathcal{A}$ can be decomposed

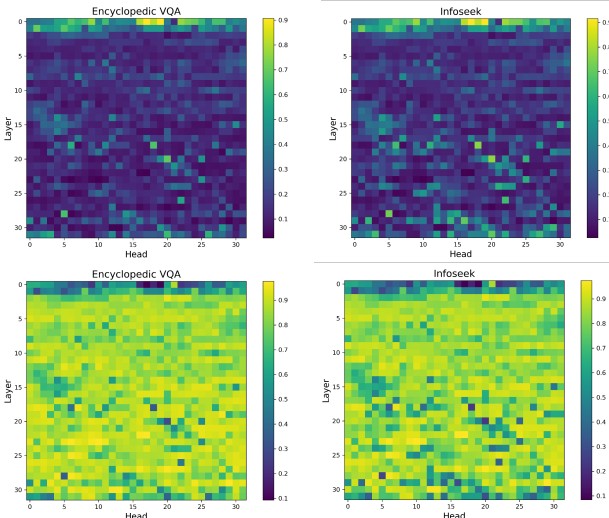

*Figure 3.* Attention Distribution across Layers and Heads. We visualize the mean attention value allocated to **visual** tokens **(top row)** and *textual* tokens **(bottom row)**, averaged over 100 random samples from **E-VQA (left column)** and **InfoSeek (right column)**. The x-axis shows attention heads, and the y-axis shows layers.

as:

$$P(\mathcal{A} \mid \mathcal{I}_Q, \mathcal{Q}, \mathcal{K}) \propto \underbrace{P(\hat{T} \mid \mathcal{I}_Q, \mathcal{Q}, \mathcal{K})}_{\text{Context Selection}} \times \underbrace{P(\mathcal{A} \mid \mathcal{I}_Q, \mathcal{Q}, \hat{T})}_{\text{Conditional Inference}},$$

(2)

where $\hat{T}$ is the selected contexts. For each query, the knowledge base $\mathcal{K}$ provides a context $T^\star$ that is sufficient to answer the question, such that the probability of the correct answer $\mathcal{A}^\star$ is significantly maximized when conditioned on $T^\star$ compared to any irrelevant context $\hat{T} \neq T^\star$:

$$P(\mathcal{A}^\star \mid \mathcal{I}_Q, \mathcal{Q}, T^\star) > P(\mathcal{A}^\star \mid \mathcal{I}_Q, \mathcal{Q}, \hat{T} \neq T^\star). \quad (3)$$

This identifies *Context Selection* as a critical factor, as the accuracy of the final answer can be strongly influenced by the quality of the selected context.

Previous approaches generally retrieves the candidate images $\hat{\mathcal{I}}$ based on the query image and question. The textual contexts associated with these images are then reranked to select $\hat{T}$. Formally, the selected context is computed as

$$\hat{T} = \arg\max_T P(T \mid \mathcal{I}_Q, \mathcal{Q}, \hat{\mathcal{C}}), \quad (4)$$

$$\hat{\mathcal{C}} = \bigcup_{I \in \hat{\mathcal{I}}} \mathcal{T}_{e(I)}, \quad \hat{\mathcal{I}} \sim P(\hat{\mathcal{I}} \mid \mathcal{I}_Q, \mathcal{Q}, \mathcal{K}), \quad (5)$$

where $e(I)$ maps an image to its associated entity, $\mathcal{T}_{e(I)}$ denotes the entity's textual contexts, and $\hat{\mathcal{C}}$ is the resulting candidate context pool induced by the retrieved images $\hat{\mathcal{I}}$.

In contrast, our method aims to directly retrieves candidate entities $\hat{\mathcal{E}}$ using their multimodal representations $(I_e, T_e)$,

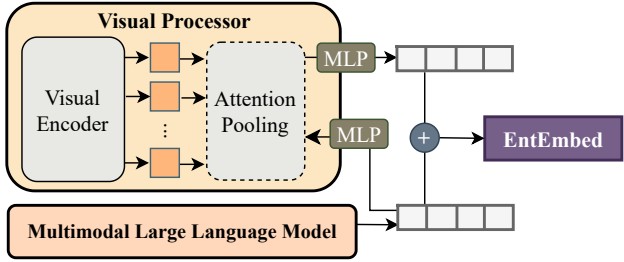

*Figure 4.* Architecture of EntBind. The MLLM first introduces an entity token, [ENT], to generate an entity-centric multimodal embedding. Visual patches are extracted using an existing visual encoder and aggregated through attention pooling that interacts with the ent token embedding to condition on entity features. The pooled visual embedding is projected by an MLP and fused with the ent token embedding to produce the final representation.

and selects the context $\hat{T}$ from the associated textual snippets:

$$\hat{T} = \arg\max_T P(T \mid \mathcal{I}_Q, \mathcal{Q}, \hat{\mathcal{E}}), \quad (6)$$

$$\hat{\mathcal{E}} \sim P(\hat{\mathcal{E}} \mid \mathcal{I}_Q, \mathcal{Q}, \mathcal{K}). \quad (7)$$

Motivated by the observation that visual and textual information often capture different aspects of an entity, we aim to learn multimodal entity representations for which jointly leveraging visual and textual evidence yields stronger discriminative signals for entity relevance than using either modality alone:

$$P\Big(e \mid \mathcal{I}_Q, \mathcal{Q}, (I_e, T_e)\Big) > \max\Big\{ P(e \mid \mathcal{I}_Q, \mathcal{Q}, I_e),$$
$$P(e \mid \mathcal{I}_Q, \mathcal{Q}, T_e)\Big\}. \quad (8)$$

Based on analysis in Appendix A, this rationale leads to a more effective retrieval strategy at the entity level, providing stronger support for selecting optimal context.

## 4. Method

In this section, we introduce EntRAG, a RAG framework for KB-VQA, as illustrated in Figure 5. The pipeline first performs entity retrieval (Section 4.1), then refines results through entity-context reranking (Section 4.2), and finally generates context-aware answers (Section 4.3).

### 4.1. Entity Retrieval

Accurate entity grounding requires representations that integrate both visual and textual information, motivating the use of multimodal embeddings. MLLM-based models, such as MM-Embed (Lin et al., 2025) have demonstrated strong performance in learning such representations. Nevertheless,

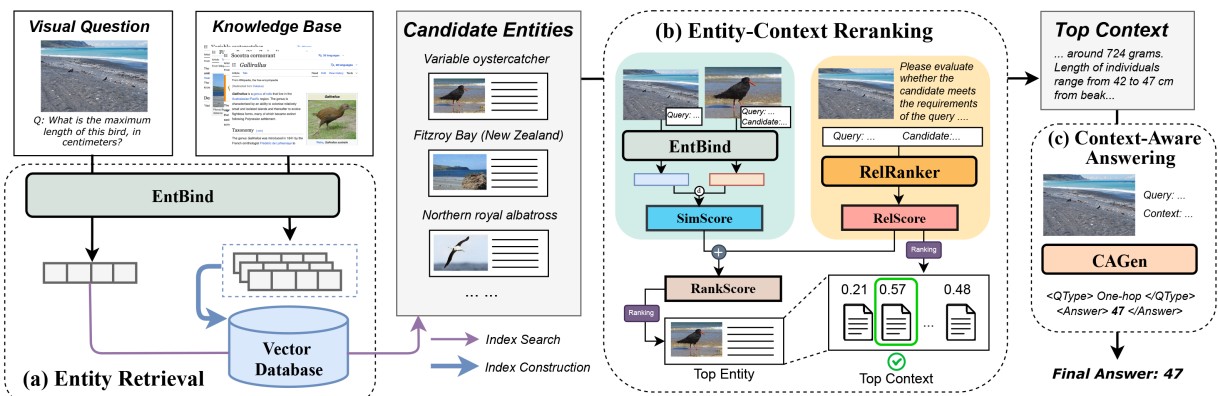

*Figure 5.* Overview of the EntRAG framework. The system consists of three stages: (a) Entity Retrieval: EntBind computes query embeddings and retrieves top-$k$ candidate entities from a vector-indexed knowledge base. (b) Entity–Context Reranking: Candidate entities are reranked by combining the similarity score from EntBind with the probabilistic relevance score from RelRanker, after which the most relevant context within each entity is selected. (c) Context-Aware Answering: The top context is provided to a fine-tuned MLLM, CAGen, to generate the final structured answer.

in practice, we observe limitations in MLLM's ability to preserve fine-grained visual information.

To quantify the role of visual features in MLLM-driven embeddings, we analyze visual and textual token attention across layers and heads in MM-Embed (Lin et al., 2025), using 100 visual–question pairs from E-VQA and InfoSeek. As illustrated in Figure 3, visual attention is prominent in early layers but collapses in later stages, whereas attention to textual tokens remains dominant throughout the final layers. Consequently, the final embeddings are largely textual, losing fine-grained visual cues critical for distinguishing similar entities. This behavior is illustrated in Figure 2, where MM-Embed emphasizes textual context, such as the bird mentioned in the query, while underutilizing visual details that differentiate entities.

To address aforementioned limitations, we propose a model that operates from two perspectives: (i) leveraging MLLMs to extract entity-relevant information; (ii) conditioning visual feature extraction on language cues to explicitly emphasize image features that are most informative for fine-grained entity identification. The overall architecture is illustrated in Figure 4.

We firstly introduce a `[ENT]` token into the MLLM vocabulary as a dedicated token to encode entity-specific embeddings. Given a query $(\mathcal{I}_Q, \mathcal{Q})$ and a candidate entity item $(I_e, T_e)$, the model is prompted with instruction $\mathcal{S}_Q$ and $\mathcal{S}_e$ to derive entity-relevant attributes from the query and candidate item respectively. We extract the final hidden states corresponding to `[ENT]` token to form the query

token embedding $\mathbf{z_q^t}$ and candidate token embedding $\mathbf{z_e^t}$:

$$\mathbf{z_q^t} = f_{\theta_t}(\mathcal{I}_Q, \mathcal{Q}, \mathcal{S}_Q), \qquad (9)$$
$$\mathbf{z_e^t} = f_{\theta_t}(I_e, T_e, \mathcal{S}_e), \qquad (10)$$

where $\theta_t$ denotes the MLLM parameters.

To further enrich the visual representation, we implement a visual processor that utilizes linguistic cues to re-weight visual features. Let $\{x_1, \cdots, x_N\} = \mathcal{V}(I)$ be the patch-level features from a visual encoder $\mathcal{V}$. We treat the projected `[ENT]` token embedding $\mathbf{y_c^t} = MLP(\mathbf{z_c^t})$ as a query to attend over the visual patches, to derive the visual embedding as:

$$\mathbf{z_c^v} = \mathbf{W}_o \left( \sum_{i=1}^{N} \text{softmax}_i \left( \frac{(\mathbf{W}_q \mathbf{y_c^t})^\top (\mathbf{W}_k \mathbf{x}_i)}{\sqrt{D}} \right) \mathbf{x}_i \right), \qquad (11)$$

where $\mathbf{W}_q, \mathbf{W}_k, \mathbf{W}_o$ are learnable projection matrices, and $c \in \{q, e\}$ indicates that the same process is applied to both the query and candidate entities.

The final entity-centric embedding $\mathbf{z_c}$ is constructed via a fusion of `[ENT]` token embedding and the visual features:

$$\mathbf{z_c} = \mathbf{z_c^t} + MLP(\mathbf{z_c^v}). \qquad (12)$$

To align the representations, we optimize the model using a InfoNCE contrastive objective, which maximizes the similarity between the query and the positive candidate entity:

$$\mathcal{L}_{ent} = -\log \frac{\exp(\mathbf{z_q} \cdot \mathbf{z_e^+}/\tau)}{\sum_{e \in \mathcal{K}} \exp(\mathbf{z_q} \cdot \mathbf{z_e}/\tau)}. \qquad (13)$$

## 4.2. Entity-Context Reranking

In this section, we introduce ECRanker, which serves two purposes: (1) to locate the most relevant entity, and (2) to select the supporting context that provides evidence for answering the question. Our method relies on two complementary scores as follows.

**Query-Candidate Similarity.** To determine whether a candidate entity matches the query $(\mathcal{I}_Q, \mathcal{Q})$, we compute a query-conditioned embedding and define the entity-level similarity as the maximum similarity over all its image–text pairs:

$$\mathbf{z_{e_i}} = f_\theta(I_e^i, T_e^i, \mathcal{Q}, \mathcal{S}_{qc}), \tag{14}$$

$$s_{\text{sim}}(e) = \max_i \langle \mathbf{z_{e_i}}, \mathbf{z_q} \rangle, \tag{15}$$

where $\mathbf{z}_q$ is computed in the same way as described in Section 4.1.

**Likelihood-based Context Relevance.** To capture the evidentiary value of a context, we measure its alignment with the query $(\mathcal{I}_Q, \mathcal{Q})$ using a likelihood-based relevance score computed from an MLLM, denoted as RelRanker. Following (Liang et al., 2022), we formulate relevance assessment as a constrained binary classification task. The model is prompted to judge whether a given context answers the query, and the posterior probability assigned to the "Yes" token is used as a continuous relevance score.

$$s_{rel}^i = P_{\theta_r}(yes|\mathcal{I}_Q, \mathcal{Q}, T_e^i, \mathcal{S}_{rel}), \tag{16}$$

where $\mathcal{S}_{rel}$ denotes the instruction template for relevance assessment and $\theta_r$ represents the model parameters. The model is trained using a standard binary cross-entropy loss over yes/no labels:

$$\mathcal{L}_{rel} = -\sum_{i=1}^{N}(y_i \log P_{\theta_r}(\text{yes}|\mathcal{I}_Q, \mathcal{Q}, T_e^i) \tag{17}$$
$$+ (1 - y^i) \log P_{\theta_r}(\text{no}|\mathcal{I}_Q, \mathcal{Q}, T_e^i)),$$

where $y_i$ represents the ground-truth label for sample $i$. Similarly, we define the entity-level relevance as the maximum relevance across all associated contexts:

$$s_{\text{rel}}(e) = \max_i s_{rel}(T_e^i). \tag{18}$$

We integrate the entity-level similarity and relevance scores to obtain a unified measure:

$$s_{\text{score}}(e) = s_{\text{sim}}(e) + s_{\text{rel}}(e), \tag{19}$$

$$\hat{e} = \arg\max_{e \in \mathcal{E}} s_{\text{score}}(e), \tag{20}$$

where $\hat{e}$ denotes the entity that best matches the query.

Upon identifying the optimal entity $\hat{e}$, we perform a refinement step to extract the most informative evidence for question answering. Specifically, we select the top-ranked context $\hat{T}$ as the one with the highest likelihood-based relevance among all contexts associated with $\hat{e}$:

$$\hat{T} = \arg\max_{T_{\hat{e}}^i \in \mathcal{T}_{\hat{e}}} s_{rel}^i. \tag{21}$$

## 4.3. Context-Aware Answer Generation

We finetune a MLLM $\theta_a$ to generate a structured response $\mathcal{A}$. To address multi-hop questions, we augment the model's vocabulary with specialized tokens: [QTYPE] for question classification, [ANS] for the terminal answer, and [NEXTQ] for sub-question decomposition in multi-hop scenarios. The generation process is formulated as:

$$\mathcal{A} = f_\theta(\hat{T}, \mathcal{I}_Q, \mathcal{Q}, \mathcal{S}_{\text{gen}}) \tag{22}$$

where $\mathcal{S}_{\text{gen}}$ is a prompt that enforces the structured output format for question answering.

For single-hop queries, the model generates the question type and the final answer directly. For multi-hop queries requiring sequential evidence retrieval, the model emits the [NEXTQ] token followed by a generated sub-query.

## 5. Experiments

In this section, we present the datasets and main experimental results. Additional implementation details and further experimental results are provided in Appendix B and Appendix D respectively.

### 5.1. Dataset

We evaluate our proposed method on two KB-VQA benchmarks: Encyclopedic VQA (E-VQA) (Mensink et al., 2023) and InfoSeek (Chen et al., 2023).

**Encyclopedic VQA.** The E-VQA dataset comprises 221k unique question-answer pairs. Each query is associated with up to five candidate images sourced from the iNaturalist 2021 (Van Horn et al., 2021) and Google Landmarks v2 (Weyand et al., 2020) datasets. The benchmark includes both single-hop and complex two-hop questions, necessitating multi-step reasoning. The supporting knowledge base consists of approximately 2M Wikipedia entities. Following the official split, we utilize 1M training pairs, 13.6k validation items, and 5.8k test items.

**InfoSeek.** InfoSeek is a large-scale dataset featuring 1.3M image-question pairs, partitioned into 934k training, 73k validation, and 348k testing instances. Since ground-truth labels for the test set are not publicly available, we conduct our primary evaluation on the validation set, consistent with established literature. Adopting the experimental protocol from (Cocchi et al., 2025; Yan & Xie, 2024), we utilize a curated knowledge base subset containing 100k entities.

*Table 1.* Entity retrieval performance on Encyclopedic-VQA. I and T denote image and text modalities.

| Model | Modality | E-VQA | | |
|---|---|---|---|---|
| | | R@1 | R@5 | R@20 |
| CLIP ViT-L/14 | I→T | 1.4 | 1.9 | 2.3 |
| CLIP ViT-L/14 | I→I | 11.4 | 22.5 | 33.2 |
| EVA-CLIP-8B | I→T | 0.9 | 2.0 | 2.5 |
| EVA-CLIP-8B | I→I | 20.0 | 36.4 | 50.5 |
| EVA-CLIP-18B | I→T | 3.9 | 7.9 | 11.3 |
| EVA-CLIP-18B | I→I | 22.8 | 40.0 | 53.4 |
| MM-Embed | IT → IT | 11.9 | 23.4 | 36.7 |
| EntBind | IT → IT | **24.1** | **43.6** | **58.3** |

*Table 2.* Entity Retrieval performance on Infoseek.

| Model | Modality | Infoseek | | |
|---|---|---|---|---|
| | | R@1 | R@5 | R@20 |
| CLIP ViT-L/14 | I→T | 13.4 | 27.8 | 41.4 |
| CLIP ViT-L/14 | I→I | 32.5 | 52.9 | 66.0 |
| EVA-CLIP-8B | I→T | 5.8 | 9.2 | 12.1 |
| EVA-CLIP-8B | I→I | 51.0 | 70.3 | 78.9 |
| EVA-CLIP-18B | I→T | 5.9 | 10.7 | 12.6 |
| EVA-CLIP-18B | I→I | 52.8 | 71.9 | 80.6 |
| MM-Embed | IT → IT | 36.7 | 60.9 | 77.3 |
| EntBind | IT → IT | **58.5** | **78.4** | **87.5** |

## 5.2. Results

### 5.2.1. ENTITY RETRIEVAL RESULTS

We report the entity retrieval performance on EVQA and InfoSeek in Table 1 and Table 2, respectively. EntBind consistently outperforms existing baselines, achieving a 1.3% improvement on EVQA and a 5.7% gain on InfoSeek for Top-1 Recall. To compare with the properties of multimodal encoders, we include image-to-text and image-to-image configurations. We find that EVA-CLIP consistently excels in image-to-image tasks, suggesting that high-fidelity visual features are indispensable for precise visual entity recognition. Notably, while MM-Embed incorporates both modalities, its heavy reliance on language signals hinders the capture of fine-grained visual cues, leading to suboptimal entity recognition compared to EVA-CLIP. In contrast, our approach effectively fuses granular visual signals with entity-relevant token embedding, demonstrating improved discriminative performance.

### 5.2.2. VQA RESULTS

We present our main VQA results in Table Table 3. Our approach consistently outperforms recent competitive baselines. Specifically, our model surpasses the highest baselines by around 10% on E-VQA and 2% on InfoSeek. These

improvements underscore the importance of entity-centric approach for KB-VQA.

### 5.2.3. ABLATION STUDY OF ENTBIND

Table 4 displays the ablation study for EntBind. We evaluate our architectural choices across four configurations: (1) **Impact of Visual Granularity.** Replacing our visual processor with a standard CLIP backbone results in a drop of over 10% in R@1, indicating that coarse-grained CLIP features are inadequate for fine-grained entity discrimination. (2) **Attention vs. Adaptive Pooling.** Substituting entity-guided attention pooling with standard learned-weight pooling leads to a 3.6% decline in R@1, confirming that entity-conditioned aggregation is more effective at capturing relevant multimodal features. (3) **Impact of Visual Embedding.** Removing the visual processor entirely causes a performance collapse exceeding 30%, demonstrating that explicit visual modeling is essential for accurate entity grounding. Notably, the remaining entity token embedding still outperforms MM-Embed, validating its discriminative capacity. (4) **Impact of Token Embedding.** Eliminating the entity token representation reduces the model to adaptive pooling over visual patch features, yielding performance similar to EVA-CLIP-8B and underscoring the critical role of entity token guidance in aligning visual evidence with entity semantics.

### 5.2.4. ABLATION STUDY OF ECRANKER

We conduct an ablation study on ECRanker to evaluate its effectiveness across two granularities: entity-level and context-level. In the entity-level evaluation, a retrieval is considered successful if the top-ranked item corresponds to the correct entity. In the context-level evaluation, both the entity identification and the specific textual snippet must match the ground truth. The results are displayed in Table 5, where we compare with EchoSight and ReflectiVA.

At the entity level, ECRanker outperforms the next best model by about 15%. The results suggest that similarity scores and relevance scores together provide complementary signals that, when combined, substantially improve entity grounding.

To analyze how entity identification influences context retrieval, we compare three ranking strategies: *Entity-First*, which uses the top context for each top-ranked entity; *Entity-Pool*, which mixes the top five entity contexts and ranks them at the context level; and *Joint-Score*, which ranks candidates based on the sum of the similarity and relevance scores. As shown in Table 5, while EntBind performs poorly at this level, this is expected as it is optimized for entity-level identification rather than fine-grained context selection. The *Entity-First* strategy achieves the highest Top-1 recall, reinforcing our core hypothesis: anchoring the search in a verified entity is more reliable for precise context ranking than

*Table 3.* VQA accuracy on E-VQA and InfoSeek.

| Method | Model | Retrieval Mode | | E-VQA | | InfoSeek | | |
|---|---|---|---|---|---|---|---|---|
| | | | | Single-Hop | All | Unseen-Q | Unseen-E | All |
| *Zero-shot MLLMs* | | | | | | | | |
| BLIP-2 (Li et al., 2023) | Flan-T5$_{XL}$ | - | | 12.6 | 12.4 | 12.7 | 12.3 | 12.5 |
| InstructBLIP (Dai et al., 2023) | Flan-T5$_{XL}$ | - | | 11.9 | 12.0 | 8.9 | 7.4 | 8.1 |
| LLaVA-v1.5 (Liu et al., 2024) | Vicuna-7B | - | | 16.3 | 16.9 | 9.6 | 9.4 | 9.5 |
| GPT-4V (Achiam et al., 2023) | - | - | | 26.9 | 28.1 | 15.0 | 14.3 | 14.6 |
| *Retrieval-Augmented Models* | | | | | | | | |
| DPR$_{V+T}$ (Lerner et al., 2024) | Multi-passage BERT | CLIP ViT-B/32 | Visual+Textual | 29.1 | - | - | - | 12.4 |
| RORA-VLM (Qi et al., 2024) | Vicuna-7B | CLIP+Google Search | Visual+Textual | - | 20.3 | 25.1 | 27.3 | - |
| Wiki-LLaVA (Caffagni et al., 2024) | Vicuna-7B | CLIP ViT-L/14+Contriever | Textual | 17.7 | 20.3 | 30.1 | 27.8 | 28.9 |
| EchoSight (Yan & Xie, 2024) | Mistral-7B/LLaMA-3-8B | EVA-CLIP-8B | Visual | 19.4 | - | - | - | 27.7 |
| MMKB-RAG (Ling et al., 2025) | Qwen2-7B | EVA-CLIP-8B | Visual | 39.7 | 35.9 | 36.4 | 36.3 | 36.4 |
| ReflectiVA (Cocchi et al., 2025) | LLaVA-v1.5-7B | EVA-CLIP-8B | Textual | 40.6 | 39.7 | 40.4 | 39.8 | 40.1 |
| ReflectiVA (Cocchi et al., 2025) | LLaVA-v1.5-7B | EVA-CLIP-8B | Visual | 35.5 | 35.5 | 28.6 | 28.1 | 28.3 |
| VLM-PRF (Gu et al., 2026) | Qwen-2.5VL-7B | EVA-CLIP-8B | Visual+tool | 37.1 | 36.0 | 43.3 | 42.7 | 42.8 |
| VLM-PRF (Gu et al., 2026) | InternVL3-8B | EVA-CLIP-8B | Visual+tool | 40.1 | 39.2 | 43.5 | 42.1 | 42.5 |
| **EntRAG(Ours)** | LLaVA-v1.5-7B | EntBind | Visual+Textual | 48.8 | 44.3 | 39.7 | 40.5 | 40.3 |
| **EntRAG(Ours)** | LLaVA-v1.6-7B | EntBind | Visual+Textual | 48.4 | 45.2 | **44.7** | 43.5 | 43.8 |
| **EntRAG(Ours)** | Qwen-2.5-7B | EntBind | Visual+Textual | **50.5** | **46.1** | 42.1 | **45.3** | **44.5** |

*Table 4.* Ablation study of EntBind on E-VQA.

| Method | R@1 | R@5 | R@20 |
|---|---|---|---|
| EVA-CLIP-8B | 52.4 | 60.8 | 73.4 |
| MM-Embed | 1.2 | 5.6 | 10.8 |
| **EntBind** | **57.0** | **72.0** | **85.2** |
| *CLIP as Visual Encoder* | 42.4 | 56.8 | 71.0 |
| *w/o Attention Pooling* | 53.4 | 70.8 | 81.2 |
| *w/o Visual Embedding* | 21.0 | 42.4 | 61.6 |
| *w/o Token Embedding* | 51.4 | 61.0 | 70.6 |

*Table 5.* Ablation study of ECRanker on E-VQA. SimS. denotes the Similarity Score, and RelS. denotes the Relevance Score.

| Method | SimS. | RelS. | R@1 | R@5 |
|---|---|---|---|---|
| *Entity-level* | | | | |
| EchoSight | ✓ | | 56.8 | 75.4 |
| ReflectiVA | | ✓ | 62.2 | 83.0 |
| **ECRanker** | ✓ | ✓ | **78.0** | **86.6** |
| *EntBind* | ✓ | | 60.2 | 67.8 |
| *RelRanker* | | ✓ | 62.8 | 81.4 |
| *Context-level* | | | | |
| EchoSight | ✓ | | 40.4 | 66.2 |
| ReflectiVA | | ✓ | 54.8 | 78.4 |
| **ECRanker** (*Entity-First*) | ✓ | ✓ | **68.0** | 77.4 |
| **ECRanker** (*Entity-Pool*) | ✓ | ✓ | 59.4 | 81.2 |
| **ECRanker** (*Joint-Score*) | ✓ | ✓ | 62.3 | **82.2** |
| *EntBind* | ✓ | | 6.8 | 25.6 |
| *RelRanker* | | ✓ | 57.6 | 77.8 |

*Table 6.* Answering performance with oracle retrieval.

| Method | Model | VQA-Accuracy |
|---|---|---|
| Vanilla | LLaVA-v1.6-7B | 78.1 |
| ReflectiVA | LLaVA-v1.5-7B | 79.6 |
| **CAGen(Ours)** | LLaVA-v1.6-7B | **88.7** |
| **CAGen(Ours)** | Qwen-2.5VL-7B | 88.4 |

searching through an unconstrained pool of context snippets. The *Joint-Score* strategy also yields competitive results by incorporating entity signals into the ranking. Conversely, while *Entity-Pool* successfully captures the correct entity within its broader candidate set, the final ranking within that set can be slightly diverted by context-level noise, leading to lower top-1 accuracy compared to the *Entity-First* approach.

#### 5.2.5. ANSWERING WITH ORACLE RETRIEVAL

To evaluate the upper-bound performance of CAGen, we perform experiments using oracle context on the E-VQA dataset, which provides ground-truth knowledge chunks for each question–image pair. As shown in Table 6, CAGen achieves substantially higher accuracy than ReflectiVA and vanilla models, with an improvement of approximately 9%. This result indicates that our model can more effectively leverage retrieved knowledge to generate accurate answers compared to existing methods.

## 6. Conclusion

In this study, we address the challenge of KB-VQA involving long-tail knowledge by introducing EntRAG, an entity-centric RAG framework that first narrows down relevant

entities then grounds contextual information to answer questions. To improve entity recognition, we propose EntBind, which jointly leverages entity-level textual features and fine-grained visual cues. Our ECRanker further integrates entity- and context-level scores to accurately identify the target entity and retrieve the most relevant context. Finally, CAGen generates responses conditioned on the retrieved context. Extensive experiments demonstrate that EntRAG achieves strong performance across existing KB-VQA benchmarks.

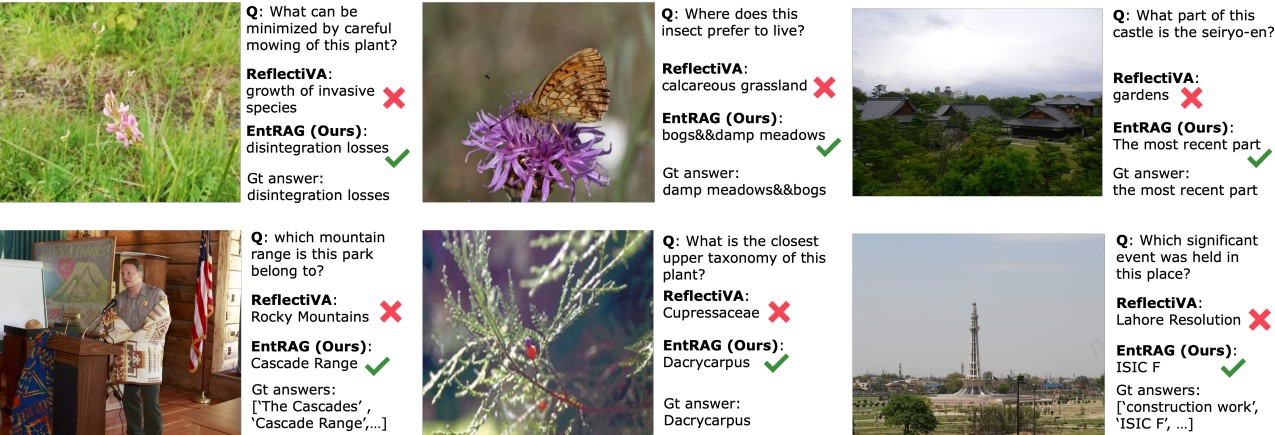

*Figure 6.* Qualitative results of visual question answering produced by EntRAG and ReflectiVA (Cocchi et al., 2025). The top row shows examples from E-VQA, while the bottom row corresponds to InfoSeek.

## Acknowledgements

Xiaoyang Wang is supported by the Australian Research Council DP240101322 and DP260100689.

## Impact Statement

This paper presents work whose goal is to advance the field of retrieval-augmented generation, specifically for Knowledge-based Visual Question Answering. EntRAG introduces an entity-centric retrieval-augmented generation framework that improves the ability of AI systems to answer questions requiring fine-grained visual and factual grounding over large-scale knowledge bases.

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

# A. Theoretical Analysis for Entity-Centric Strategy

We develop a formal theoretical argument motivating an entity-centric, multimodal retrieval strategy. The core claim is an *upper bound argument*: the class of multimodal scoring functions strictly contains the class of unimodal (image-only) scoring functions, which implies a strictly higher theoretical ceiling on correct-entity retrieval probability. This motivates jointly leveraging visual and textual representations without overclaiming guaranteed improvements in every practical setting.

## A.1. Setup and Notation

Let $\mathcal{K} = \{(I_e, T_e)\}_{e \in \mathcal{E}}$ denote the knowledge base, where each entity $e \in \mathcal{E}$ is associated with a visual representation $I_e$ (a canonical image) and a textual representation $T_e$ (a descriptive passage). Given a query $(\mathcal{I}_Q, \mathcal{Q})$ consisting of a query image $\mathcal{I}_Q$ and a natural-language question $\mathcal{Q}$, let $e^\star \in \mathcal{E}$ denote the *ground-truth entity* whose associated context $T^\star := T_{e^\star}$ is needed to answer the query.

A retrieval scoring function assigns a scalar relevance score to each entity $e$ given the query:

$$f \colon (\mathcal{I}_Q,\, \mathcal{Q},\, I_e,\, T_e) \longmapsto \mathbb{R}.$$

Given $f$, the top-$K$ retrieved entity set is

$$\hat{\mathcal{E}}_f = \underset{e \in \mathcal{E},\, |\hat{\mathcal{E}}|=K}{\arg \text{top-}K}\; f(\mathcal{I}_Q, \mathcal{Q}, I_e, T_e).$$

The final retrieved context is then selected from the textual passages of the candidate entities:

$$\hat{T} = \arg \max_{T \in \mathcal{T}_{\hat{\mathcal{E}}_f}} P(T \mid \mathcal{I}_Q, \mathcal{Q}, \hat{\mathcal{E}}_f), \quad \mathcal{T}_{\hat{\mathcal{E}}_f} := \bigcup_{e \in \hat{\mathcal{E}}_f} \{T_e\}.$$

We study two function classes:

$$\mathcal{F}_{\text{uni}} := \big\{ f \mid f(\mathcal{I}_Q, \mathcal{Q}, I_e, T_e) = g(\mathcal{I}_Q, I_e) \text{ for some } g \big\}, \tag{23}$$

$$\mathcal{F}_{\text{multi}} := \big\{ f \mid f(\mathcal{I}_Q, \mathcal{Q}, I_e, T_e) \text{ depends jointly on } (\mathcal{I}_Q, \mathcal{Q}, I_e, T_e) \big\}. \tag{24}$$

Any $f \in \mathcal{F}_{\text{uni}}$ is recovered from $\mathcal{F}_{\text{multi}}$ by setting the dependence on $(\mathcal{Q}, T_e)$ to a constant, so $\mathcal{F}_{\text{uni}} \subsetneq \mathcal{F}_{\text{multi}}$.

## A.2. Upper Bound via Function-Class Containment

Define the *oracle retrieval probability* achievable by a function class $\mathcal{F}$ as

$$R^\star(\mathcal{F}) := \sup_{f \in \mathcal{F}} P\big(e^\star \in \hat{\mathcal{E}}_f\big).$$

**Proposition A.1** (Strict Containment Implies Strictly Higher Ceiling). *$R^\star(\mathcal{F}_{\text{uni}}) \leq R^\star(\mathcal{F}_{\text{multi}})$. Moreover, the inequality is strict whenever there exists a query $(\mathcal{I}_Q, \mathcal{Q})$ for which no $f \in \mathcal{F}_{\text{uni}}$ achieves $P(e^\star \in \hat{\mathcal{E}}_f) = 1$ but some $f \in \mathcal{F}_{\text{multi}}$ does.*

*Proof.* Because $\mathcal{F}_{\text{uni}} \subsetneq \mathcal{F}_{\text{multi}}$, we have

$$R^\star(\mathcal{F}_{\text{uni}}) = \sup_{f \in \mathcal{F}_{\text{uni}}} P(e^\star \in \hat{\mathcal{E}}_f) \leq \sup_{f \in \mathcal{F}_{\text{multi}}} P(e^\star \in \hat{\mathcal{E}}_f) = R^\star(\mathcal{F}_{\text{multi}}).$$

For strictness, suppose there exists a query for which visual similarity alone is ambiguous—i.e., $P(e^\star \in \hat{\mathcal{E}}_f) < 1$ for all $f \in \mathcal{F}_{\text{uni}}$—yet the textual representation $T_{e^\star}$ contains discriminative information not encoded in $I_{e^\star}$. Then the multimodal scorer $f^*(\mathcal{I}_Q, \mathcal{Q}, I_e, T_e) = g(\mathcal{I}_Q, I_e) + h(\mathcal{Q}, T_e)$, for appropriately chosen $g$ and $h$, achieves $P(e^\star \in \hat{\mathcal{E}}_{f^*}) > P(e^\star \in \hat{\mathcal{E}}_f)$ for any $f \in \mathcal{F}_{\text{uni}}$. $\square$

## A.3. Multimodal Complementarity

The strict gap in Proposition A.1 is non-vacuous precisely when visual and textual modalities provide complementary discriminative signals. We formalize this as follows.

**Assumption A.2** (Modality Complementarity). For any entity $e$, let $\epsilon_I(e)$ and $\epsilon_T(e)$ denote the retrieval errors induced by image-only and text-only scoring, respectively. We assume $\epsilon_I(e)$ and $\epsilon_T(e)$ are *not perfectly correlated*: there exist entities $e \neq e^\star$ for which $\epsilon_I(e^\star) > 0$ (the image scorer fails) but $\epsilon_T(e^\star) = 0$ (the text scorer succeeds), and vice versa.

Under Assumption A.2, a joint scoring function that combines both modalities can exploit cases where one modality is ambiguous by deferring to the other, strictly reducing the overall retrieval error.

## A.4. Connecting Retrieval to Context Accuracy

Let $\hat{T}$ denote the context selected by the reranking step over $\mathcal{T}_{\hat{\mathcal{E}}_f}$. We have the following chain of inequalities.

**Corollary A.3** (End-to-End Context Accuracy). *Under the information bottleneck assumption that $T^\star \in \mathcal{T}_{\hat{\mathcal{E}}_f}$ is necessary for $\hat{T} = T^\star$, and letting $f^\star_{\mathrm{uni}} \in \mathcal{F}_{\mathrm{uni}}$ and $f^\star_{\mathrm{multi}} \in \mathcal{F}_{\mathrm{multi}}$ denote the respective oracle scorers,*

$$P(\hat{T} = T^\star \mid f^\star_{\mathrm{uni}}) \;\leq\; P\Big(e^\star \in \hat{\mathcal{E}}_{f^\star_{\mathrm{uni}}}\Big) \;<\; R^\star(\mathcal{F}_{\mathrm{multi}}) \;\leq\; P(\hat{T} = T^\star \mid f^\star_{\mathrm{multi}}).$$

*Proof.* The first inequality follows from the information bottleneck: $\hat{T} = T^\star$ requires $e^\star \in \hat{\mathcal{E}}_f$ as a necessary condition, so $P(\hat{T} = T^\star) \leq P(e^\star \in \hat{\mathcal{E}}_f)$. The strict middle inequality is Proposition A.1 under Assumption A.2. The final inequality follows by definition of $R^\star$ and the fact that correct entity retrieval is sufficient for correct context selection when reranking is accurate. $\qquad\square$

## A.5. Noise Decomposition and Complementary Signal

For a candidate entity $e$, each modality produces a noisy relevance score:

$$s_I(e) = \phi_I(e) + \eta_I(e), \qquad s_T(e) = \phi_T(e) + \eta_T(e), \tag{25}$$

where $\phi_I(e), \phi_T(e)$ are true discriminative signals and $\eta_I(e), \eta_T(e)$ are modality-specific noise terms. Naive fusion $s_I(e) + s_T(e)$ accumulates both noise terms and can degrade retrieval when they are positively correlated.

However, the two modalities capture *distinct* aspects of entity identity. Decomposing the true signals as $\phi_I(e) = \phi_{\mathrm{sh}}(e) + \phi_I^\perp(e)$ and $\phi_T(e) = \phi_{\mathrm{sh}}(e) + \phi_T^\perp(e)$, where $\phi_{\mathrm{sh}}$ is shared and $\phi_I^\perp, \phi_T^\perp$ are modality-exclusive components, the *complementary gain* is

$$\Delta_{\mathrm{comp}}(e) := \phi_I^\perp(e) + \phi_T^\perp(e), \tag{26}$$

which is strictly positive for $e^\star$ under Assumption A.2. The ideal scorer $f^*(e) = \phi_I(e) + \phi_T(e)$ then achieves a strictly larger pairwise margin over any distractor $e \neq e^\star$ compared to any $f \in \mathcal{F}_{\mathrm{uni}}$:

$$f^*(e^\star) - f^*(e) \;\geq\; \phi_I(e^\star) - \phi_I(e), \tag{27}$$

with strict inequality when either modality contributes discriminative signal not captured by the other. Our model approximates $f^*$ by jointly encoding visual-textual signals, encouraging the learned representations to capture the complementary signal $\Delta_{\mathrm{comp}}$ and maximize the margin in Eq. (27).

## A.6. Discussion

Corollary A.3 establishes that the theoretical ceiling on end-to-end context accuracy is strictly higher for $\mathcal{F}_{\mathrm{multi}}$ than for $\mathcal{F}_{\mathrm{uni}}$, with the gap driven by the complementary gain $\Delta_{\mathrm{comp}}(e^\star) = \phi_I^\perp(e^\star) + \phi_T^\perp(e^\star)$ whenever the two modalities carry non-redundant discriminative signal. Crucially, this upper bound does not assert that every multimodal model outperforms every unimodal model in practice, since a poorly trained scorer may accumulate correlated noise and degrade retrieval through modality misalignment. Rather, it shows that the capacity to improve exists uniquely within $\mathcal{F}_{\mathrm{multi}}$, providing theoretical motivation for designing entity-centric retrievers that jointly leverage both visual representations $I_e$ and textual representations $T_e$. In practice, we adopt contrastive learning to realize this potential by isolating modality-exclusive components, promoting cross-modal consistency, and penalizing entities with contradictory visual–textual signals.

# B. Experiments Setup

## B.1. Evaluation Metrics

Following the official protocols of each benchmark, we report Recall@$k$ to assess retrieval effectiveness. For question answering, we employ dataset-specific metrics to account for varying response formats. For E-VQA, correctness is determined via a BERT-based semantic matching score between the predicted and ground-truth answers to accommodate linguistic variation. For InfoSeek, we report the VQA Score, which applies type-specific heuristics to normalize accuracy across different answer categories (e.g., numeric, boolean, or entity-based).

## B.2. Implementation

We adopt LLaVA-v1.6-7B as the multimodal large language model (MLLM) backbone funless otherwise specified. The knowledge base is organized at the entity level, where each entity is associated with multiple images and textual snippets. For textual snippets without a corresponding image, we randomly sample a visual instance from the same entity to form a multimodal context. At the answer generation stage, the model first predicts the question type for each input. For single-hop questions, the answer is generated directly. For multi-hop questions, the model first generates an intermediate sub-question (denoted [NEXTQ]), which is then used as a new query and passed through the same retrieval and reranking pipeline to produce the final answer. Following the EVQA setting, we limit reasoning to a maximum of two hops.

The EntBind is trained using EVA-CLIP-8B as the backbone, due to its strong capability in capturing fine-grained visual semantics. During training, positive pairs are constructed from query–context pairs referring to the same entity, while contexts from different entities are treated as negatives. For efficient large-scale retrieval, the resulting embeddings are indexed using Faiss, and at inference time we retrieve the top-$k$ candidate entities based on Euclidean distance. For the reranking stage, we further fine-tune EntBind by incorporating the query into the context encoder, enabling query-conditioned representation learning. To enhance discriminative capability, we mine hard negatives based on errors from the initial retrieval stage. Both the RelRanker and CAGen modules are trained using Low-Rank Adaptation (LoRA) with rank $r = 32$ and scaling factor $\alpha = 64$. These modules are optimized for two epochs with a learning rate of $2 \times 10^{-5}$. During inference, we rerank the top 100 retrieved entities for E-VQA and the top 30 retrieved entities for InfoSeek, reflecting differences in knowledge base scale.

For training, we jointly utilize the E-VQA and InfoSeek datasets. Since InfoSeek does not provide an official test split, we randomly partition its training data into training and validation sets, ensuring that entities do not overlap across splits.

For ablation studies on both E-VQA and InfoSeek, we randomly sample 500 query instances. We sample about 50K entities for E-VQA and 5K entities for InfoSeek based on their knowledge base scales. The statistics are summarized in Table 7

*Table 7.* Subset statistics for ablation experiments.

| Dataset | # Queries | Knowledge Base Size (By Entity) |
|---------|-----------|---------------------------------|
| EVQA    | 500       | 49,822                          |
| Infoseek| 500       | 4,836                           |

## B.3. Comparison of Parameter Settings and Ensemble Strategies

Below we provide a comprehensive breakdown of the parameter settings across all compared methods:

*Table 8.* Parameter and model overview across compared methods. [†] denotes reranking, [‡] denotes answering, and [†‡] denotes both.

| Method | Retrieval | Reranking & Answering |
|--------|-----------|------------------------|
| Wiki-LLaVA | CLIP ViT-L/14 + Contriever (∼550M) | Vicuna-7B[‡] or LLaMA-3.1-8B[‡] |
| EchoSight | EVA-CLIP-8B | BLIP-2 (∼100M)[†], Mistral-7B[‡] or LLaMA3-8B[‡] |
| MMKB-RAG | EVA-CLIP-8B | Qwen-2-7B[†‡] |
| ReflectiVA | EVA-CLIP-8B | LLaVA-v1.5-7B[†‡] |
| VLM-PRF | EVA-CLIP-8B | Qwen-2.5VL-7B[†‡] |
| **Ours** | EntiBind (∼15B) | LLaVA-v1.6-7B[†], LLaVA-v1.6-7B[‡] |

Prior methods (e.g., ReflectiVA, VLM-PRF) rely on a single model to jointly rerank and generate answers, assuming that context scoring aligns with the reasoning required for answering, which ensures reasoning consistency. In contrast, our method separates these stages into (1) integrating entity-level (similarity score) and context-level reranking (relevance score), and (2) answer generation conditioned on the final selected evidence and hop-type detection. This structural separation motivates the fine-tuning of different models for each task.

## C. Prompts

This section details the prompts used across different stages of the framework. Specifically, we include:

- $\mathcal{S}_Q$: prompts used by EntBind to encode queries during entity retrieval (see Figure 7);

- $\mathcal{S}_e$: prompts used by EntBind to encode contextual evidence during entity retrieval (see Figure 8);

- $\mathcal{S}_{qc}$: prompts used by EntBind for query-conditioned context encoding during the reranking stage (see Figure 9);

- $\mathcal{S}_{rel}$: prompts used by RelRanker to assess relevance between queries and candidate contexts (see Figure 10);

- $\mathcal{S}_{gen}$: prompts used by CAGen for context-aware answer generation (see Figure 11).

```
User: [IMAGE]
Given the query and the image, extract the information relevant to the main entity and
encode it into the entity token.
Query: [Query]
Entity Token: [ENT]
```

*Figure 7.* Prompt of $\mathcal{S}_Q$.

```
User: [IMAGE]
Given the context and the image, extract the information relevant to the main entity and
encode it into the entity token.
Context: [Query]
Entity Token: [ENT]
```

*Figure 8.* Prompt of $\mathcal{S}_e$.

```
User: [IMAGE]
Given the context and the image, extract the information relevant to the query entity and
 encode it into the entity token.
Query: [Query]
Context: [Context]
Entity Token: [ENT]
```

*Figure 9.* Prompt of $\mathcal{S}_{qc}$.

## D. Experimental Results

### D.1. Reranking Performance under Varying Top-$k$ Candidates

Table 9 shows the effect of varying the number of top-$k$ retrieved candidates on reranking performance. As $k$ increases, recall consistently improves across all metrics, indicating that including more candidates provides the reranker with additional relevant entities to consider.

### D.2. Effects of Reranking Components

We extend the original reranking formulation in Eq. 20 by explicitly weighting the contribution of each score component:

$$s_{\text{score}}(e) = \alpha \, s_{\text{rel}}(e) + \beta \, s_{\text{sim}}(e), \tag{28}$$

```
User: [IMAGE]
I will provide you with a query and a candidate. Please evaluate whether the candidate
meets the requirements of the query, and respond with 'Yes' or 'No'.
Query: [Query]
Candidate: [Context]
```

*Figure 10.* Prompt of $\mathcal{S}_{rel}$.

```
User: [IMAGE]
Given an image-text query and supporting context, identify the hop type and produce the
first-hop question and answer, and generate the next subquestion if the type is multi-hop
.
Query: [Query]
Supporting Context: [Context]
```

*Figure 11.* Prompt of $\mathcal{S}_{gen}$.

where $s_{\mathrm{rel}}(e)$ is the likelihood-based relevance score from RelRanker, $s_{\mathrm{sim}}(e)$ is the similarity score from EntBind, and $\alpha, \beta$ are weighting coefficients controlling their relative contributions. To evaluate the contribution of each component, we vary one weighting coefficient while fixing the other to 1 and evaluate recall@k. As shown in Figure 12, on EVQA, setting both weights to 1 consistently yields the most stable and strongest performance across different values of $k$. Increasing either coefficient beyond 1 leads to a small degradation in recall. On InfoSeek, performance exhibits a small bias toward the similarity score; nevertheless, the balanced setting ($\alpha = \beta = 1$) remains competitive, achieving robust performance. The overall behavior indicates that the relevance and similarity scores provide complementary information and are most effective when combined with comparable importance.

### D.3. Ablation Study of EntBind on Infoseek.

Table 10 presents an ablation study of EntBind on the Infoseek dataset. EntBind consistently achieves the highest recall across all metrics compared to baseline methods, demonstrating the effectiveness of its multimodal entity representations. These results align with the observations in Section 5.2.3, reinforcing the importance of integrating both visual and textual information. Among the ablations, removing the visual embedding results in the largest performance drop, which highlights the critical role of visual information for accurate entity recognition. Removing token embeddings or attention pooling also reduces recall, though to a lesser extent. It indicate that textual cues and the attention mechanism contribute complementary evidence that enhances overall retrieval.

*Table 9.* Reranking performance of EntBind under different top-$k$ retrievals.

| Retrieval | R@1 | R@5 | R@20 | R@50 |
|---|---|---|---|---|
| Top-5 | 34.7 | - | - | - |
| Top-20 | 44.4 | 52.1 | - | - |
| Top-50 | 49.6 | 59.7 | 63.6 | - |
| Top-100 | 51.5 | 63.1 | 67.7 | 70.1 |

*Table 10.* Ablation study of EntBind on Infoseek

| Method | R@1 | R@5 | R@20 |
|---|---|---|---|
| EVA-CLIP-8B | 61.8 | 69.7 | 73.0 |
| MM-Embed | 4.4 | 14.4 | 32.8 |
| **EntBind** | **65.6** | **71.8** | **77.8** |
| *CLIP as Visual Encoder* | 50.4 | 60.6 | 66.8 |
| *w/o Attention Pooling* | 64.2 | 70.0 | 75.4 |
| *w/o Visual Embedding* | 30.2 | 41.4 | 58.0 |
| *w/o Token Embedding* | 62.7 | 69.8 | 75.0 |

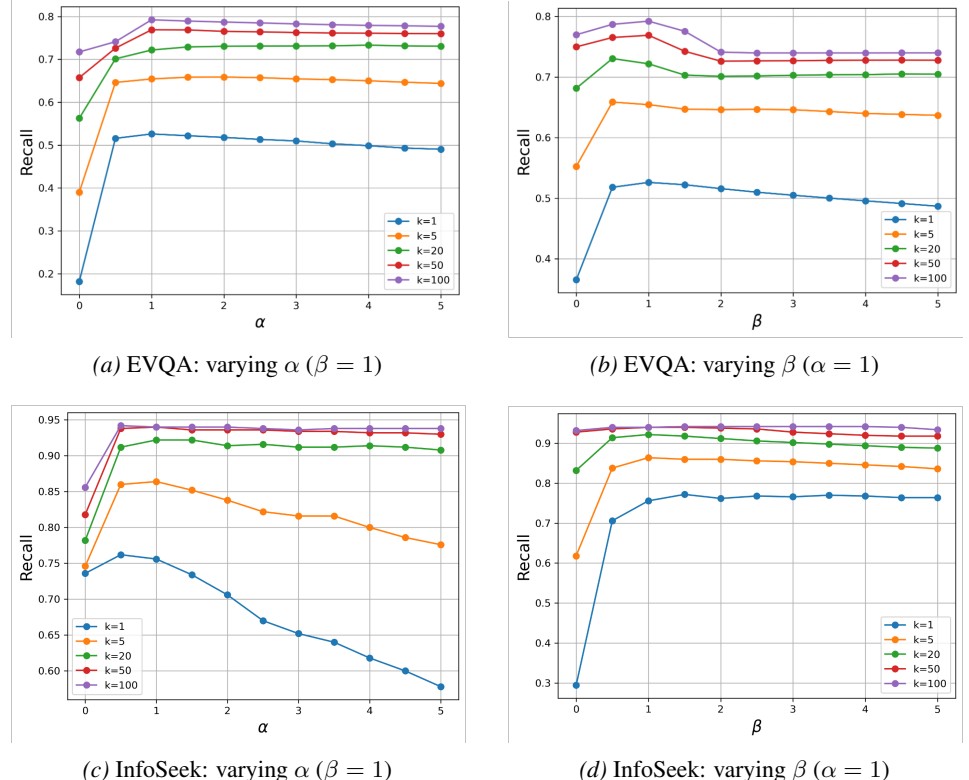

*(a)* EVQA: varying $\alpha$ $(\beta = 1)$  *(b)* EVQA: varying $\beta$ $(\alpha = 1)$

*(c)* InfoSeek: varying $\alpha$ $(\beta = 1)$  *(d)* InfoSeek: varying $\beta$ $(\alpha = 1)$

*Figure 12.* Effect of varying the weighting coefficients $\alpha$ and $\beta$ on reranking performance (recall@k). Top row: EVQA. Bottom row: InfoSeek. Left column varies $\alpha$ with $\beta = 1$ fixed; right column varies $\beta$ with $\alpha = 1$ fixed.

## D.4. Ablation Study of ECRanker on Infoseek

Table 11 demonstrates the ablation study of ECRanker on Infoseek. Since Infoseek does not provide ground-truth context chunks, we report entity-level retrieval results only. The results demonstrate that utilizing both scores in ECRanker substantially improves recall@1 and recall@5 compared to using either score individually. In particular, using only the similarity score or only the relevance score leads to significantly lower recall, while the full ECRanker model effectively integrates these complementary signals to achieve the highest retrieval performance. It further confirms that both entity-level similarity and contextual relevance are essential for accurate multimodal retrieval.

*Table 11.* Ablation study of ECRanker on Infoseek. SimS. denotes the Similarity Score, and RelS. denotes the Relevance Score.

| Method | SimS. | RelS. | R@1 | R@5 |
|---|---|---|---|---|
| *Entity-level* | | | | |
| EchoSight | ✓ | | 56.7 | 80.6 |
| ReflectiVA | | ✓ | 57.2 | 85.8 |
| **ECRanker** | ✓ | ✓ | **79.1** | **87.6** |
| *EntBind* | ✓ | | 75.6 | 83.2 |
| *RelRanker* | | ✓ | 51.6 | 75.8 |

## D.5. Inference Time

We report the inference time of each stage: retrieval, reranking, and answer generation, in Table 12. Compared to the baselines, EntRAG is relatively slower, particularly in the retrieval and reranking stages, due to its entity-focused processing. However, the framework remains practical, with efficient answer generation and clear gains in accuracy, demonstrating a reasonable trade-off between performance and speed.

*Table 12.* Inference time of models at each stage. We measure the runtime over 500 examples and report the average.

| Model | Retrieval | Reranking | Answering |
|---|---|---|---|
| EchoSight | | 0.009s | 0.545s |
| ReflectiVA | 0.013s | 0.138s | 0.132s |
| EntRAG (Ours) | 0.068s | 0.120s | 0.469s |

## D.6. Qualitative Results

Figure 13 visualizes the embeddings using t-SNE. We show embeddings generated by EntBind, including both the visual embedding and token embedding. The combined embedding effectively brings the positive examples closest to the query, demonstrating that integrating visual-level and token-level information produces a more discriminative representation for entity retrieval.

## D.7. Qualitative Results of Failure Cases

Figure 14 presents qualitative examples of failure cases from our method. We observe that most errors stem from incorrect entity or context selection. These cases highlight the importance of robust context selection in multimodal question answering.

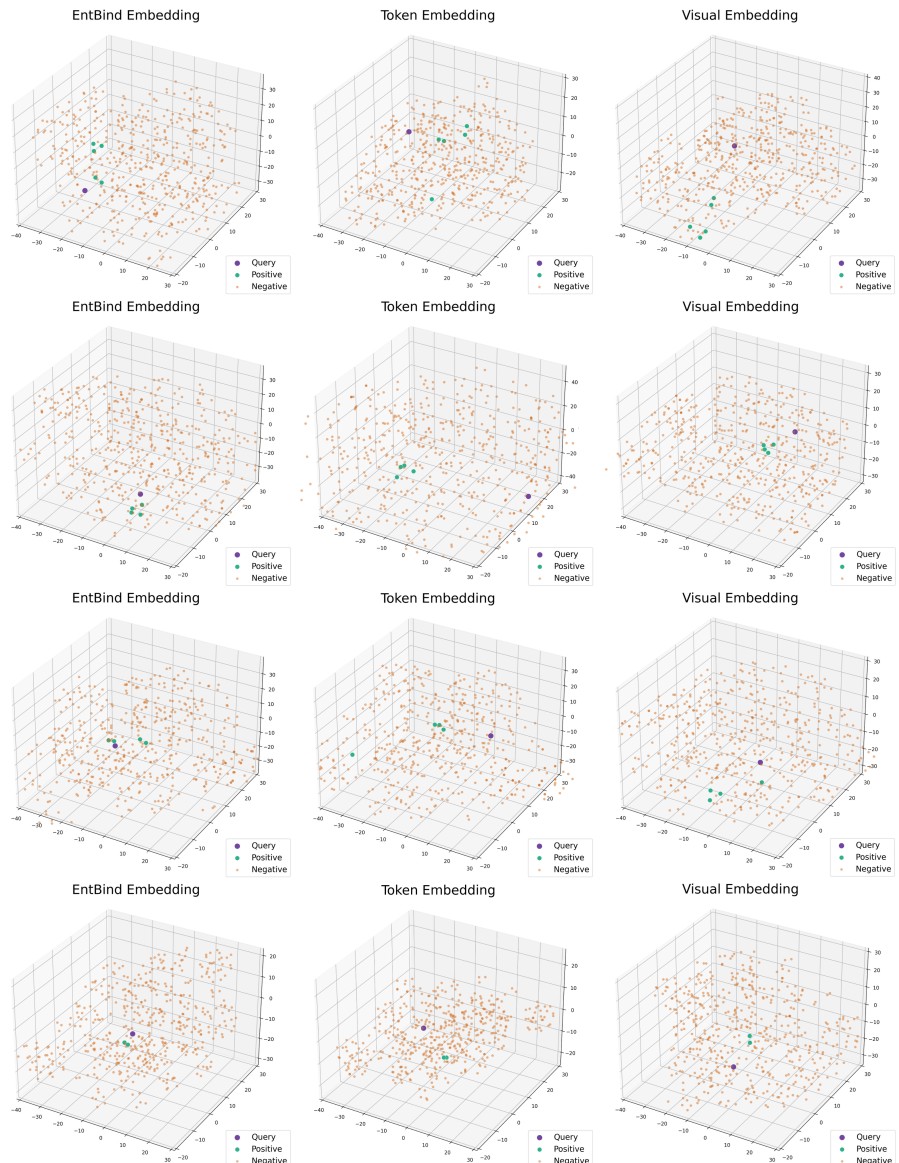

*Figure 13.* t-SNE visualization of embeddings produced by EntBind. The top two rows correspond to examples from E-VQA, and the bottom two rows correspond to examples from Infoseek.

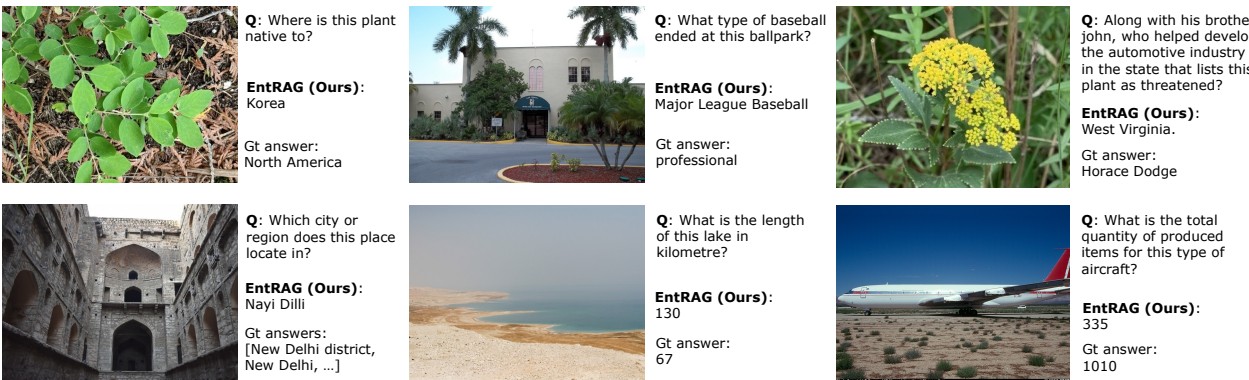

*Figure 14.* Qualitative failure cases of our method. The top row shows examples from E-VQA, and the bottom row is from Infoseek.

