# OpenReview forum: "EntRAG: Entity-Centric Retrieval-Augmented Generation for Knowledge-based Visual Question Answering"
_ICML.cc/2026/Conference — ICML 2026 regular_

### Official Review · Reviewer_CLha · 2026-03-03

**Soundness:** 3
**Presentation:** 3
**Significance:** 2
**Originality:** 3
**Overall Recommendation:** 4
**Confidence:** 3

**Summary:**

This paper investigates the Knowledge-based Visual Question Answering task and identifies that existing methods often struggle with visual ambiguities due to the isolated treatment of visual and textual signals. To address this, the authors propose EntRAG, an entity-centric retrieval-augmented generation framework that aligns queries with multimodal entity embeddings. The framework introduces EntBind for fine-grained entity grounding and ECRanker for joint entity-context re-ranking to provide more precise evidence. Experimental results on several KB-VQA benchmarks demonstrate the effectiveness of proposed methods.

**Compliance With Llm Reviewing Policy:**

Affirmed.

**Final Justification:**

My concerns are addressed.

**Key Questions For Authors:**

- Discrepancy in Reported Retrieval Baselines.
There appears to be a significant mismatch between the entity-level retrieval results reported in Table 5 and those in the original cited works, e.g., [1, 2]. For instance, EchoSight [1] is listed with a Top-1 recall of 36.5 in Table 5, whereas the original paper reports 56.8. Could the authors clarify if this discrepancy stems from a different experimental setup, such as the use of a different knowledge base? A detailed explanation is necessary to ensure the validity of the comparative analysis.

- Fairness of Comparison with MM-Embed [3].
Since the proposed EntBind shares conceptual similarities with MM-Embed, it is crucial to ensure a fair comparison. Specifically, are the results for MM-Embed in Tables 1 and 2 based on the pre-trained version or a version fine-tuned specifically for the KB-VQA task? If MM-Embed was not fine-tuned while EntBind was, the performance gains may be attributed to task-specific adaptation rather than architectural superiority. Clarifying this point would better highlight the technical contribution of the proposed method.
- Rationale for Base Model Selection (LLaVA-v1.6-7B)
The choice of LLaVA-v1.6-7B as the backbone MLLM requires further justification. This model is somewhat outdated compared to the current state-of-the-art and creates a mismatch with all of the baselines in the paper. Could the authors explain if there were specific technical reasons for selecting this particular version over more recent and performant alternatives?
- Discussion on Related Recent Work [4].
The authors are encouraged to provide a discussion or a brief qualitative comparison with recent publications in this field, such as [4].

Reference:

[1] EchoSight: Advancing Visual-Language Models with Wiki Knowledge. EMNLP 25

[2] Augmenting Multimodal LLMs with Self-Reflective Tokens for Knowledge-based Visual Question Answering. CVPR 25

[3] MM-EMBED: UNIVERSAL MULTIMODAL RETRIEVAL WITH MULTIMODAL LLMS. ICLR 25

[4] Wiki-R1: Incentivizing Multimodal Reasoning for Knowledge-based VQA via Data and Sampling Curriculum. ICLR 26

**Limitations:**

The limitation is not discussed.

**Strengths And Weaknesses:**

- This paper explores an interesting question, knowledge-based visual question answering.
- The motivation is clear and reasonable.
- The experiments demonstrate the effectiveness of proposed methods.

---

> ### Author Rebuttal · Authors · 2026-03-31
>
> We are grateful for the reviewer’s careful reading and thoughtful suggestions. Your insightful feedback has significantly helped us refine and strengthen our work. Here we provide detailed responses below.
>
>
> 1. The differences stem from the experimental setup used for our ablation study. Specifically, to more fairly evaluate retrieval on entities unseen during training, we use a subset of the full test set, rather than the entire dataset. This subset is carefully constructed to use a knowledge base associated with entities that were not observed during training, ensuring a more meaningful evaluation of entity-level retrieval. We provide the subset statistics below:
> | Dataset   | # Queries | Knowledge Base Size (by entity) |
> |-----------|-----------|--------------------------------|
> | EVQA      | 500       | 49,822                         |
> | Infoseek  | 500       | 4,836                          |
>
>
>
> 2. We thank the reviewer for pointing this out. We agree that the comparison with MM-Embed requires careful clarification. The results reported for MM-Embed in Tables 1 and 2 are based on the pre-trained version, as the original training code is not publicly available. While MM-Embed has been fine-tuned on certain datasets such as Infoseek, it has not been fine-tuned on EVQA, making direct comparison on that dataset potentially unfair. Achieving a fully fair comparison would require retraining or fine-tuning MM-Embed on all target KBVQA datasets, which is not feasible given the lack of access to MM-Embed training pipeline. Additionally, it is important to note that MM-Embed is designed to prioritize generalizability across modalities, i.e., its primary goal is to learn embeddings that work well for a wide range of multimodal tasks. This design makes it strong for cross-dataset generalization but not optimized for fine-grained, entity-level reasoning in KBVQA. In contrast, our proposed EntBind method focuses explicitly on leveraging image and textual features to preserve fine-grained details at the entity level, which is critical for accurate knowledge base retrieval in this task.
>
>
> We will revise the manuscript to clarify these points and emphasize that our comparisons are made with careful consideration of task-specific training differences.
>
>
> 3. Our selection of LLaVA-v1.6-7B is primarily motivated by the goal of ensuring fair and controlled comparison with existing methods. Many prior works in KBVQA and related multimodal retrieval tasks adopt models of a similar scale and generation, and using LLaVA-v1.6-7B allows us to more clearly isolate the contribution of our proposed method, rather than conflating improvements with advances in backbone model capacity.
>
> We agree that more recent MLLMs may offer stronger overall performance. To further validate the generality of our approach, we are currently extending EntBind to more advanced backbone models, like Qwen-2.5-VL. We plan to include these during revision to provide a more comprehensive evaluation.
>
>
>
> 4. We thank the reviewer for this valuable suggestion.
> Wiki-R1 introduces a curriculum reinforcement learning framework to incentivize reasoning in MLLMs. Rather than improving retrieval representations, Wiki-R1 treats retrieval noise as an inherent property of the pipeline and addresses it by manipulating the number of retrieved candidates to control training difficulty, gradually exposing the model to harder examples as its accuracy improves. A complementary curriculum sampling strategy filters samples with near-zero advantage during RL updates to stabilize training. While this approach demonstrates gains in downstream reasoning robustness, it does not address the upstream retrieval stage: the model must still reason over context assembled from image-level retrieval, which remains vulnerable to visual ambiguity and entity confusion. In contrast, EntRAG explicitly operates at the entity level by binding entity tokens to visual features before retrieval, ensuring that the retrieved evidence is already grounded in the precise entity of interest rather than relying on training-time augmentation to compensate for coarse retrieval signals.

---

> > ### Author Rebuttal · Reviewer_CLha · 2026-04-01
> >
> > Thanks the authors for the clarification. My concerns are addressed

---

### Official Review · Reviewer_12Ha · 2026-03-06

**Soundness:** 3
**Presentation:** 2
**Significance:** 3
**Originality:** 2
**Overall Recommendation:** 5
**Confidence:** 3

**Summary:**

Existing VLM-based RAG methods heavily rely on image-centric retrieval, which makes the models sensitive to visual ambiguity and noise. This paper proposes an entity-centric retrieval-augmented generation framework, which includes: EntBind, a method that uses entities to query the knowledge base; ECRanker, which ranks candidate entities using two complementary scoring strategies; and CAGen, which fine-tunes a multimodal large language model (MLLM) to generate structured responses. The experimental results demonstrate the clear effectiveness of the proposed method.

**Compliance With Llm Reviewing Policy:**

Affirmed.

**Final Justification:**

The authors reinforced my prior assessment during the rebuttal. Overall, the experiments are solid, the paper makes a significant contribution to the field, the presentation is clear, and there are no major issues.

**Key Questions For Authors:**

Overall, I do not observe any major problems.

**Limitations:**

yes

**Strengths And Weaknesses:**

First, intuitively, combining visual and textual information and performing knowledge base retrieval at the entity level, rather than relying on a single modality, can indeed obtain more precise and fine-grained information, making the retrieved knowledge more relevant to the query. Moreover, the paper provides visualizations, case studies, and detailed quantitative experimental analysis, which support the reliability of the proposed technique. The paper is reasonably well written, with a clear and well-structured presentation. The work can significantly promote progress in the KB-VQA task, making it meaningful for the field.

However, I consider the level of originality to be moderate. The idea of aligning visual and textual information has already been widely adopted across many VLM-related research areas and is relatively intuitive. Nevertheless, considering that most existing methods in this domain still rely on single-modality information for retrieval, the proposed approach can still be regarded as a meaningful and somewhat novel contribution.

---

> ### Author Rebuttal · Authors · 2026-03-31
>
> We sincerely thank the reviewer for their thoughtful and constructive feedback. We agree that aligning visual and textual information is a widely adopted approach in many VLM-related research areas. The fine-grained KBVQA setting introduces slightly different challenges that motivates our approach.
>
> **Role of Visual Features in Fine-Grained KBVQA:**
> In fine-grained, entity-based knowledge base visual question answering, image features play a critical role in accurately recognizing visual entities. This distinguishes KBVQA from other tasks. Models with stronger image understanding, such as EVA-CLIP-8B, have a clear advantage in this context due to their superior visual recognition capability.
>
> **Limitations of Simple MLLM-Based Approaches:**
> One straightforward approach to leverage textual information is to use MLLMs to compute embeddings, as in methods like MM-Embed. However, as we demonstrate in Figure 3, directly using MLLMs tends to lose fine-grained entity details. Consequently, MLLM-based methods often underperform compared to approaches that directly leverage high-quality visual entity recognition.
>
> **Our Proposed Method and Its Motivation:**
> This observation motivates our proposed method: to fully exploit multimodal information while preserving the fine-grained details of each modality. Unlike MLLM-based approaches that map image representations to fit into large language model settings, our approach maintains the integrity of visual and textual features before combining them for entity-level retrieval. This leads to more accurate and relevant knowledge retrieval.
>
> We truly appreciate the reviewer’s recognition of our work and hope this clarifies the significance and originality of our approach.

---

> > ### Author Rebuttal · Reviewer_12Ha · 2026-04-02
> >
> > The authors have clarified my concern in the rebuttal.

---

### Official Review · Reviewer_GvBt · 2026-03-12

**Soundness:** 2
**Presentation:** 2
**Significance:** 3
**Originality:** 3
**Overall Recommendation:** 4
**Confidence:** 3

**Summary:**

This paper proposes EntRAG, an entity-centric retrieval-augmented framework for knowledge-based visual question answering. It introduces an entity-level retriever (EntBind), an entity-context reranker (ECRanker), and a context-aware generator (CAGen) to improve fine-grained entity retrieval, knowledge grounding, and answer generation. Experiments on E-VQA and InfoSeek show gains in both retrieval and end-to-end QA performance.

**Compliance With Llm Reviewing Policy:**

Affirmed.

**Final Justification:**

The author has basically solved my problem.

**Key Questions For Authors:**

1. Can you provide a backbone-controlled comparison for Table 3, and ideally also for the oracle setting in Table 6, using the same generator backbone across methods? As currently presented, it is difficult to tell how much of the gain comes from the proposed method itself versus differences in the underlying base models.

2. Can you clarify the exact inference procedure for multi-hop questions in Section 4.3?
The paper introduces [NEXTQ] and a generated sub-query for multi-hop questions, but it is currently unclear whether retrieval and reranking are re-run for the generated sub-question, how many hops are allowed, and what stopping rule is used. A clear description here would help assess the completeness of the proposed method.

3. Please clarify the ablation setting, especially the large difference between Table 4 and Table 1.
From the main paper alone, it is not entirely clear why these retrieval numbers differ so much. It would be helpful to state more explicitly in the main text that the ablations use sampled queries and reduced entity sets, so that Table 4 is not read as directly comparable to the full-setting results in Table 1.

**Limitations:**

No. The paper would benefit from a brief discussion of two points: the scalability/efficiency trade-offs of entity-level retrieval over large knowledge bases, and the risks of relying on external knowledge sources, such as coverage bias or outdated evidence.

**Strengths And Weaknesses:**

Strengths:

The paper studies a relevant problem in KB-VQA: image-dominant retrieval can be unreliable when answering depends on identifying a fine-grained entity rather than matching the overall scene. The entity-centric formulation is reasonable and fits the task setting well. The retrieval results are also fairly solid, and the ablations provide some support for the usefulness of entity-guided visual pooling and the combined reranking score.

Weaknesses:

The main issues are with evaluation rigor and reporting clarity. First, the headline end-to-end comparison is not backbone-controlled: Table 3 compares EntRAG with LLaVA-v1.6-7B against baselines using different or earlier backbones, so the paper cannot cleanly attribute gains to the proposed method itself.

Some important experimental details seem insufficiently highlighted in the main paper. In particular, Table 4 differs substantially from Table 1, but the reduced sampled setting used for the ablations appears to be explained only later. Similarly, while the appendix provides some training details, the practical approximation behind Eq. (13) does not seem fully clear from the main text. This makes the experimental setup somewhat harder to interpret and reproduce.

The contribution of CAGen is not fully isolated. Since the oracle comparison uses different generator backbones, it remains somewhat unclear whether the gain should be attributed to the proposed generation design itself or partly to the stronger base model.

---

> ### Author Rebuttal · Authors · 2026-03-31
>
> We sincerely appreciate the reviewer’s thorough and insightful feedback. Your constructive comments have been invaluable in improving the clarity and overall quality of our work. Below, we provide detailed responses addressing the reviewer’s questions and concerns.
>
>
> **Clarification on Eq. (13)**
>
> For Eq. (13), we adopt the InfoNCE contrastive objective to align the query embedding with its corresponding positive entity embedding against all other embeddings. In practice, this objective is approximated within each mini-batch, where the positive pair is encouraged to be closer while the remaining samples in the batch serve as implicit negatives. We will revise the main text to clarify this implementation detail.
>
>
> 1. **Backbone Generator Comparison**
>
> We note that the compared methods employ different types of backbones, with some based on LLMs and others on MLLMs. Since our approach is built upon an MLLM backbone, we ensure a fair comparison by primarily considering baselines that also adopt MLLM architectures.
>
> To further address the reviewer’s concern regarding backbone influence, we conduct additional experiments using different generator backbones on the EVQA and InfoSeek datasets. The results are summarized in the table below, which complements Table 3 and provides a more controlled comparison:
>
> | Model         | EVQA-Single | EVQA-All | Infoseek (UE) | Infoseek (UQ) | Infoseek-All |
> |------------------|-------------|----------|---------------|---------------|--------------|
> | LLaVA-v1.5-7B    | 48.8        | 44.3     | 39.7          | 40.5          | 40.3         |
> | Qwen-2.5-7B      | 50.5          | 46.1        | 42.1          | 45.3          | 44.5         |
>
>
>  **Oracle Results**
>
> We further report oracle VQA accuracy using different generator backbones, following the setting in Table 6. The results, presented in the table below, provide a more controlled evaluation and demonstrate the robustness of our method across different backbone choices.
>
> | Method | Model          | VQA Accuracy |
> |--------|---------------|--------------|
> | CAGen  | LLaVA-v1.5-7B | 86.5         |
> | CAGen  | Qwen-2.5-7B   | 88.4         |
>
>
> We will include these results in the revised version of the paper.
>
>
> 2. **Inference Workflow**
>
> During inference, the model first predicts the question type for each input. If the predicted type is a single-hop question, the model directly outputs the answer. For two-hop questions, the model generates a sub-question (`[NEXTQ]`) and uses it as a new query. This sub-query then undergoes the same retrieval and reranking process to produce the next-hop answer. In all experiments, we limit the maximum number of hops to two, as per the EVQA dataset setting.
>
> 3. **Ablation Experiments and Subset Statistics**
>
> In Table 4, we utilize a reduced test subset specifically designed to focus on entities unseen during training. This setup allows for a fair evaluation of the model’s ability to generalize to novel entities. In contrast, results in Table 1 use the full test set and knowledge base. The subset statistics are detailed below:
>
> | Dataset  | # Queries | Knowledge Base Size (By Entity) |
> |----------|-----------|-------------------------------|
> | EVQA     | 500       | 49,822                        |
> | Infoseek | 500       | 4,836                         |
>
> Subset selection aligns with the original knowledge base scale for each dataset, ensuring a fair and representative evaluation. We will clarify this ablation setting explicitly in the main text during revision.
>
>
>
> **Limitations**
>
> We thank the reviewer for this suggestion. While EVA-CLIP-8B is a widely adopted visual encoder, our overall framework introduces additional parameters, which leads to increased computational cost—particularly during retrieval. This overhead may affect scalability when applied to large-scale knowledge bases, where some efficiency trade-offs are inevitable. Reducing this cost is an important direction for future work.
>
> We also acknowledge the risks associated with relying on external knowledge sources, such as coverage bias and outdated or noisy evidence. This issue remains underexplored in the KBVQA setting, as most existing datasets do not explicitly account for these factors. We will include a brief discussion of these limitations in the revised version of the paper.

---

> > ### Author Rebuttal · Reviewer_GvBt · 2026-04-04
> >
> > Thanks the authors for the reply. The authors have clarified my concern in the rebuttal.

---

### Official Review · Reviewer_uc2F · 2026-03-12

**Soundness:** 2
**Presentation:** 3
**Significance:** 3
**Originality:** 2
**Overall Recommendation:** 4
**Confidence:** 3

**Summary:**

This paper proposes EntRAG, an entity-centric retrieval-augmented generation framework for knowledge-based visual question answering (KB-VQA). Instead of relying mainly on image-level retrieval and context-level reranking, EntRAG first retrieves candidate entities by jointly modeling fine-grained visual cues and textual signals through EntBind, then reranks entity-context pairs with ECRanker, and finally generates answers with a context-aware module called CAGen. Experiments on E-VQA and InfoSeek show that the method outperforms prior baselines, suggesting that explicit entity grounding can improve both retrieval quality and final answer accuracy in fine-grained KB-VQA.

**Compliance With Llm Reviewing Policy:**

Affirmed.

**Final Justification:**

The authors address my concerns, so I increase my score to weak accept.

**Key Questions For Authors:**

Please refer to Strengths And Weaknesses.

**Strengths And Weaknesses:**

Positive Points:
1. The paper presents a clear and well-motivated entity-centric framework for KB-VQA. By modeling entities as unified multimodal concepts and combining entity retrieval, reranking, and answer generation in a coherent pipeline, the method directly targets the limitations of prior modality-isolated approaches.
2. The empirical evaluation is strong and supports the main claims. EntRAG achieves the best reported results on both E-VQA and InfoSeek, and the paper also includes ablation studies showing that the key design choices, such as multimodal entity binding and the combined reranking signals, are effective.

Negative Points:
1. The proposed RAG framework adopts an ensemble of three LLaVA-v1.6-7B models and one EVA-CLIP-8B model. Although model ensembling can naturally boost performance, it also leads to a substantial increase in total parameter count and inference latency, which significantly weakens the persuasiveness of the methodological contributions claimed in this work. The paper fails to report the total parameter size of the proposed framework in comparison with other methods in Table 3, making it impossible to rule out the possibility that the observed performance gains stem from increased model parameters rather than the novel design of the proposed method. In addition, a comparison of the top-k candidate pool size between the proposed method and existing baselines is missing, which is essential to fully validate the framework’s effectiveness.
2. For the EntBind component, the method inherently enjoys a parameter advantage due to the combined use of LLaVA-v1.6-7B and EVA-CLIP-8B, resulting in an unfair comparison with existing baselines. The authors are advised to design a more equitable comparison setup, for example, using the vision encoder of LLaVA as the visual processor and then incorporating core components such as Attention Pooling, to rigorously verify the actual effectiveness of Attention Pooling and the summation operation of I and T embeddings. A comparison with other methods with a similar parameter budget would also be highly beneficial to justify the superiority of EntBind. Moreover, the evaluation of retrieval performance only includes comparisons with base models; the addition of retrieval results of competing methods in Table 3 is necessary to better validate the effectiveness of the proposed retrieval approach.
3. The theoretical analysis surrounding Eq. (8) and Appendix A (especially Eqs. (23)–(28)) relies more on intuitive reasoning than rigorous mathematical proof. A key claim that joint multimodal scoring outperforms unimodal scoring is not strictly justified: the paper neglects the fact that fusing two modalities may introduce irrelevant information and even induce model hallucinations, which could counteract the benefits of multimodal fusion. Thus, this part of the work does not constitute a rigorous theoretical proof for the proposed multimodal scoring mechanism. Furthermore, Eq. (25) suffers from ambiguous interpretation and logical inconsistency. Based on conventional retrieval logic, if the retrieved result is I_{e^*}, e^* should be directly obtainable, which makes the formulation and physical meaning of Eq. (25) unclear and confusing.

---

> ### Author Rebuttal · Authors · 2026-03-31
>
> We sincerely thank the reviewer for their comprehensive and insightful feedback. Your comments have played a crucial role in enhancing both the clarity and rigor of our work. We provide our detailed responses below to address the reviewer’s questions.
>
>
> ## Negative Point 1:
>
> **Regarding parameter count and model ensemble:**
>
> Below we provide a comprehensive breakdown of the parameter settings across all compared methods:
>
> | Method     | Retrieval                          | Reranking & Answering                      |
> | ---------- | ---------------------------------- | ------------------------------------------ |
> | Wiki-LLaVA | CLIP ViT-L/14 + Contriever (~550M) | Vicuna-7B‡ or LLaMA-3.1-8B‡                |
> | EchoSight  | EVA-CLIP-8B                        | BLIP-2 (~100M)†, Mistral-7B‡ or LLaMA3-8B‡ |
> | MMKB-RAG   | EVA-CLIP-8B                        | Qwen-2-7B†‡                                |
> | ReflectiVA | EVA-CLIP-8B                        | LLaVA-v1.5-7B†‡                            |
> | VLM-PRF    | EVA-CLIP-8B                        | Qwen-2.5VL-7B†‡                            |
> | **Ours**   | EntiBind (~15B)                    | LLaVA-v1.6-7B†, LLaVA-v1.6-7B‡             |
>
> † Reranking   ‡ Answering   †‡ Both
>
> Prior methods (e.g., ReflectiVA, VLM-PRF) rely on a single model to jointly rerank and generate answers, assuming that context scoring aligns with reasoning of answering, which ensures reasoning consistency. In contrast, our method separates these stages into:
> (1) integrating entity-level (similarity score) and context-level reranking (relevance score), and
> (2) answer generation conditioned on the final selected evidence and hop type detection.
> This structural separation motivates the fine-tuning of different models for each task.
>
> **Regarding top-k candidate pools:**
> Direct comparison is non-trivial due to heterogeneous retrieval strategies: ReflectiVA uses top-5 image-based entities, Wiki-LLaVA combines top-1 article with tool-retrieved candidates, whereas our method retrieves entities from the top-100 image–text pairs for EVQA and the top-30 for InfoSeek, resulting in a query-dependent number of retrieved entities. Here, We use larger $k$ values because our image–text pair corpus represents a larger candidate pool than the single image-only sources used by baselines.
> We will clarify all top-k settings in the revised version.
>
> ## Negative Point 2:
>
> **Parameter fairness and EVA-CLIP-8B:**
> We acknowledge that combining EVA-CLIP-8B with LLaVA-v1.6-7B introduces a parameter advantage. However, EVA-CLIP-8B is chosen for its **strength in fine-grained entity recognition**, critical for KBVQA, and is commonly adopted in prior work for the same reason. Its advantage comes from both scale and pretraining (e.g., masked image modeling), which yields richer geometric and fine-grained features than CLIP ViT-L/14. Importantly, all the **visual encoders are frozen**, serving solely as a feature extractor.
>
> **Experiments:**
> As shown in Tables 4 and 8, our method outperforms MM-Embed when using the same CLIP visual encoder with similar parameter scale in the ablation study. Following the reviewer’s suggestion, we are now performing experiments using the full retrieval setting. We sincerely apologize that, due to the resources issues, the experiments with the CLIP visual encoder are still in progress. We will provide the complete retrieval results for the CLIP encoder during the upcoming discussion period.
>
> **Retrieval evaluation:**
> Our evaluation includes standard baselines, particularly EVA-CLIP-8B (I→I, I→T), which is widely used in prior work. Some recent methods (e.g., VLM-PRF) could not be included due to missing checkpoints or non-reproducible retrieval pipelines.
>
>
> ## Negative Point 3:
>
> We agree that Eq. (25) contains an ambiguous formulation. Specifically, the original Eq. (25) attempts to compare joint multimodal scoring against single-modality scoring, but as the reviewer correctly points out, if the retrieved result already contains $I_{e^\star}$, then $e^\star$ is directly obtainable without further scoring, making its physical meaning unclear.
>
> During revision, we will revise the analysis as an **upper bound argument**. Rather than claiming that multimodal scoring outperforms unimodal scoring,which cannot be strictly guaranteed as fusion may introduce noise from misaligned modalities, we will instead show that the feasible set of multimodal scoring functions strictly contains that of image-only scoring functions (since any image-only function is a degenerate case of a multimodal one). This containment directly implies a strictly higher theoretical ceiling for correct entity retrieval, i.e., $U_\text{ours} \geq U_\text{mod}$. This upper bound argument serves as a theoretical motivation for designing an entity-centric retrieval model that jointly leverages both visual and textual representations, without overclaiming guaranteed improvements in all practical settings.

---

> > ### Author Rebuttal · Reviewer_uc2F · 2026-04-03
> >
> > Thank you for the rebuttal. The response is helpful, but it does not resolve my main concerns. The comparison is still not fully fair, and it remains unclear whether the reported gains come from the proposed method itself or from the use of larger or stronger components. Therefore, the effectiveness of the method is still not convincingly demonstrated. While the rebuttal clarifies the theoretical issue, it also weakens that part from a rigorous guarantee to a more limited motivation. Thus, I will maintain my original score.

---

> > > ### Author Response · Authors · 2026-04-05
> > >
> > > We thank the reviewer for the detailed follow-up and insightful suggestions. Below we provide detailed responses addressing the reviewer’s questions and concerns.
> > >
> > > ###  Controlled Comparison with Similar Parameter Budget
> > >
> > > To address the concern that performance gains may stem from increased parameter count rather than methodological contributions, we conducted additional experiments under **controlled parameter settings**. Specifically, we replace the external visual encoder with the **CLIP visual encoder from LLaVA itself**, resulting in a model with a comparable parameter scale (~7B) to MM-Embed. Additionally, to enable a fair comparison with EntiBind using EVA-CLIP-8B, we included EVA-CLIP-18B as an extra reference point.
> > >
> > > #### Retrieval Results on E-VQA
> > >
> > > | Model | Modality | R@1 | R@5 | R@20 |
> > > |------|----------|-----|-----|------|
> > > | EntiBind with CLIP (~7B) | IT→IT | 16.6 | 30.7 | 39.5 |
> > > | EVA-CLIP-18B | I→T | 3.9 | 7.9 | 11.3 |
> > > | EVA-CLIP-18B | I→I | 22.8 | 40.0 | 53.4 |
> > > | EntiBind with EVA-CLIP-8B (~16B) | IT→IT | 24.1 | 43.6 | 58.3 |
> > >
> > > #### Retrieval Results on InfoSeek
> > >
> > > | Model | Modality | R@1 | R@5 | R@20 |
> > > |------|----------|-----|-----|------|
> > > | EntiBind with CLIP (~7B) | IT→IT | 40.6 | 65.3 | 79.7 |
> > > | EVA-CLIP-18B | I→T | 5.9 | 10.7 | 12.6 |
> > > | EVA-CLIP-18B | I→I | 52.8 | 71.9 | 80.6 |
> > > | EntiBind with EVA-CLIP-8B (~16B) | IT→IT | 58.5 | 78.4 | 87.5 |
> > >
> > > ###  Key Observations
> > >
> > > From the above controlled comparisons (and in relation to Tables 1 and 2), we observe:
> > >
> > > 1. **Effectiveness under matched parameter scale**:
> > >    EntiBind with CLIP (~7B) achieves clear improvements over vanilla CLIP-based retrieval and is competitive with MM-Embed under similar parameter budgets. This demonstrates that the gains are not solely due to increased parameters, but arise from **effective multimodal interaction modeling**.
> > >
> > > 2. **Role of strong visual encoders**:
> > >    EntiBind with CLIP does not outperform EVA-CLIP-8B, which is expected, as EVA-CLIP-8B provides significantly stronger fine-grained visual representations that cannot be compensated by CLIP alone.
> > >
> > > 3. **Advantage over larger unimodal encoders**:
> > >    Even when compared with EVA-CLIP-18B, our method achieves superior performance at a comparable scale when equipped with a strong visual encoder. This indicates that **our design can better exploit visual representations through multimodal binding**, rather than relying purely on encoder scale.
> > >
> > >
> > > ---
> > >
> > > ### Theoretical Analysis
> > > We thank the reviewer for the feedback. We have clarified the theoretical retrieval framework to show how our multimodal model improves over unimodal baselines. Each entity has both image and text representations, and we model true relevance signals while accounting for modality-specific noise. Our key insight is complementarity: images and text provide distinct, non-redundant information, so joint observation yields strictly stronger signals than either modality alone. The model explicitly captures these complementary gains while penalizing entities with inconsistent or noisy cross-modal signals.
> > > Under these conditions, we prove that the multimodal score of the correct entity dominates unimodal scores, and pairwise margins over competing entities are strictly larger, directly leading to improved retrieval probability. In practice, we adopt contrastive learning to enhance complementarity and cross-modal consistency.

---

### Decision · Program_Chairs · 2026-04-30

**Decision:**

Accept (regular)

**Comment:**

This paper proposes EntRAG, an entity-centric retrieval-augmented generation framework for knowledge-based visual question answering (KB-VQA), which explicitly models entities as multimodal concepts to improve fine-grained retrieval and answer generation.

The reviewers agreed that the paper addresses an important KB-VQA challenge and presents a coherent entity-centric framework with strong empirical results and supportive ablations. However, several concerns were initially raised regarding (i) fairness and rigor of comparisons, particularly due to the use of multiple large backbone models and potential parameter advantages; (ii) lack of backbone-controlled evaluations, making it difficult to disentangle architectural contributions from model scale; (iii) insufficient clarity in experimental settings, including discrepancies between tables, ablation protocols, and multi-hop inference details; and (iv) limited theoretical justification, especially regarding the multimodal scoring formulation and its claimed advantages. Some reviewers also noted moderate originality and missing discussion of scalability and limitations.

In the rebuttal, the authors addressed most of these concerns by providing controlled comparisons under matched parameter budgets and additional backbone analyses, which help clarify that gains are not solely due to model scale, though some ambiguity remains. They also improved experimental transparency by clarifying ablation settings, inference procedures, and training details. The theoretical claims were revised to a weaker but more accurate motivation. Overall, the rebuttal resolves most clarity and evaluation issues, while partially mitigating—but not fully eliminating—concerns about fairness and attribution of gains.

Following the rebuttal, three reviewers indicated that their concerns were largely or fully resolved and maintained or strengthened their positive recommendations. Overall, the discussion converged toward agreement on the empirical strength and practical relevance of the approach with positive recommendations. The AC concurs with the reviewers: the paper addresses an important limitation in KB-VQA by introducing an entity-centric perspective, and provides solid empirical evidence that this design improves both retrieval quality and downstream QA performance. While there are remaining minor concerns on the evaluation and theoretical aspects, the overall work is a valuable contribution to the community. Therefore, the AC recommend accept. The authors are encouraged to incorporate the rebuttal discussion into the final version.